# EPHA4 signaling dysregulation links abnormal locomotion and the development of idiopathic scoliosis

**Lianlei Wang**[1,2,3,4†], **Xinyu Yang**[1,4†], **Sen Zhao**[1,5†], **Pengfei Zheng**[6†], **Wen Wen**[1,2,3], **Kexin Xu**[1,2,3], **Xi Cheng**[1,2,3], **Qing Li**[1,2,3], **Anas M Khanshour**[7], **Yoshinao Koike**[8,9], **Junjun Liu**[6], **Xin Fan**[10], **Nao Otomo**[8,9], **Zefu Chen**[1,2,3], **Yaqi Li**[1,2,3], **Lulu Li**[11], **Haibo Xie**[6], **Panpan Zhu**[6], **Xiaoxin Li**[1,2,12], **Yuchen Niu**[1,2,12], **Shengru Wang**[1,2,3], **Sen Liu**[1,2,3], **Suomao Yuan**[4], **Chikashi Terao**[8], **Ziquan Li**[1,2,3], **Shaoke Chen**[10], **Xiuli Zhao**[13], **Pengfei Liu**[5,14], **Jennifer E Posey**[14], **Zhihong Wu**[1,2,3,12], **Guixing Qiu**[1,2,3,12], **DISCO study group (Deciphering Disorders Involving Scoliosis & COmorbidities)**, **Shiro Ikegawa**[9], **James R Lupski**[15,16,17], **Jonathan J Rios**[7,18,19], **Carol A Wise**[7,18,19], **Jianguo T Zhang**[1,2,3\*], **Chengtian Zhao**[6\*], **Nan Wu**[1,2,3\*]

\*For correspondence:
jgzhang_pumch@yahoo.com
(JTZ);
chengtian_zhao@ouc.edu.cn
(CZ);
dr.wunan@pumch.cn (NW)

†These authors contributed
equally to this work

Competing interest: See page
21

Reviewing Editor: Bruce Appel,
University of Colorado School of
Medicine, Aurora, United States

[1]Department of Orthopedic Surgery, State Key Laboratory of Complex Severe and Rare Diseases, Peking Union Medical College Hospital, Peking Union Medical College and Chinese Academy of Medical Sciences, Beijing, China; [2]Beijing Key Laboratory of Big Data Innovation and Application for Skeletal Health Medical Care, Beijing, China; [3]Key Laboratory of Big Data for Spinal Deformities, Chinese Academy of Medical Sciences, Beijing, China; [4]Department of Orthopaedic Surgery, Qilu Hospital of Shandong University, Cheeloo College of Medicine, Shandong University, Jinan, China; [5]Department of Molecular and Human Genetics, Baylor College of Medicine, Houston, United States; [6]Institute of Evolution & Marine Biodiversity, College of Marine Life Science, Ocean University of China, Qingdao, China; [7]Center for Pediatric Bone Biology and Translational Research, Scottish Rite for Children, Dallas, United States; [8]Laboratory for Statistical and Translational Genetics, RIKEN Center for Integrative Medical Sciences, Yokohama, Japan; [9]Laboratory for Bone and Joint Diseases, RIKEN Center for Integrative Medical Sciences, Yokohama, Japan; [10]Department of Pediatric Endocrine and Metabolism, Maternal and Child Health Hospital of Guangxi Zhuang Autonomous Region, Nanning, China; [11]Department of Newborn Screening Center, Beijing Obstetrics and Gynecology Hospital, Capital Medical University, Beijing Maternal and Child Health Care Hospital, Beijing, China; [12]Department of Central Laboratory, Peking Union Medical College Hospital, Peking Union Medical College and Chinese Academy of Medical Sciences, Beijing, China; [13]Department of Medical Genetics, Institute of Basic Medical Sciences, Chinese Academy of Medical Sciences and Peking Union Medical College, Beijing, China; [14]Baylor Genetics, Houston, United States; [15]Departments of Pediatrics, Texas Children's Hospital and Baylor College of Medicine, Houston, United States; [16]Texas Children's Hospital, Houston, United States; [17]Human Genome Sequencing Center, Baylor College of Medicine, Houston, United States; [18]Department of Orthopaedics, University of Texas Southwestern Medical Center, Dallas, United States; [19]McDermott Center for Human Growth and Development, University of Texas Southwestern Medical Center, Dallas, United States

## eLife Assessment

Genetic variants have been strongly implicated in idiopathic scoliosis (IS), however, the list of variants that are causative of IS is not complete and the cellular and molecular mechanisms that underlie IS are poorly understood. These authors combined human genetic analysis with zebrafish experiments to produce **valuable** evidence that alleles that impair function of EPHA4 cause IS, thereby extending our understanding of the basis of IS. The human genetic data are quite **convincing** but the zebrafish work lacks some validations and details.

**Abstract** Idiopathic scoliosis (IS) is the most common form of spinal deformity with unclear pathogenesis. In this study, we first reanalyzed the loci associated with IS, drawing upon previous studies. Subsequently, we mapped these loci to candidate genes using either location-based or function-based strategies. To further substantiate our findings, we verified the enrichment of variants within these candidate genes across several large IS cohorts encompassing Chinese, East Asian, and European populations. Consequently, we identified variants in the *EPHA4* gene as compelling candidates for IS. To confirm their pathogenicity, we generated zebrafish mutants of *epha4a*. Remarkably, the zebrafish *epha4a* mutants exhibited pronounced scoliosis during later stages of development, effectively recapitulating the IS phenotype. We observed that the *epha4a* mutants displayed defects in left-right coordination during locomotion, which arose from disorganized neural activation in these mutants. Our subsequent experiments indicated that the disruption of the central pattern generator (CPG) network, characterized by abnormal axon guidance of spinal cord interneurons, contributed to the disorganization observed in the mutants. Moreover, when knocked down *efnb3b*, the ligand for Epha4a, we observed similar CPG defects and disrupted left-right locomotion. These findings suggested that ephrin B3-Epha4 signaling is vital for the proper functioning of CPGs, and defects in this pathway could lead to scoliosis in zebrafish. Furthermore, we identified two cases of IS in *NGEF*, a downstream molecule in the EPHA4 pathway. Collectively, our data provide compelling evidence that neural patterning impairments and disruptions in CPGs may underlie the pathogenesis of IS.

## Introduction

Idiopathic scoliosis (IS) is the most common form of spinal deformity, affecting about 2.5% of the global population (*Hresko, 2013*; *Luk et al., 2010*). IS may have long-term physical and mental health consequences, such as cosmetic deformity, cardiopulmonary impairment, and even disability (*Weinstein et al., 2003*). All these consequences can severely reduce the quality of life. Early intervention with conservative treatments, such as braces, can control scoliosis progression and reduce the need for surgical intervention (*Weinstein et al., 2013*). However, IS often remains undiagnosed until malformation is evident, emphasizing the importance of risk-prediction measurements in medical management.

Genetic factors are thought to play a significant role in the development of IS, while only a few genes have been associated with the condition to date (*Cheng et al., 2015*; *Miller, 2007*). Several single-nucleotide polymorphisms (SNPs) associated with susceptibility to IS have been identified through genome-wide association studies (GWASs), including SNPs linked to genes such as *LBX1*, *GPR126*, and *BNC2*. Notably, knockdown or overexpression of zebrafish homologs of these IS-associated genes has yielded body axis defects (*Guo et al., 2016*; *Kou et al., 2019*; *Kou et al., 2013*; *Ogura et al., 2015*; *Zhu et al., 2015*). In addition to common SNPs, rare variants with larger effect sizes or causing rare Mendelian disease traits also contribute to IS. Rare variants in *FBN1*, *FBN2*, and other extracellular matrix genes are associated with severe IS (*Buchan et al., 2014*; *Haller et al., 2016*). Linkage analysis of familial IS suggests that it follows autosomal dominant (AD), X-linked dominant, and multifactorial patterns of inheritance (*Miller et al., 2005*). Variants in *CHD7* and *AKAP2* are also implicated in the pathogenesis of Mendelian forms of IS (*Gao et al., 2007*; *Liebeskind et al., 2016*). Finally, centriolar protein POC5 and planar cell polarity protein Vang-like protein 1 (VANGL1) were also shown to be associated with IS. Notably, all these genes participate in a wide range of biological processes, and the mechanisms underlying IS progression still require further investigation. Consequently, there is currently no consensus on the etiology of IS (*Tang et al., 2021*).

In this study, we developed a novel pipeline that combines SNP-to-gene mapping and rare variant association analysis. Using this approach, we analyzed a large Chinese population with IS and further validated our findings in East Asian and European populations. Through these analyses, we identified *EPHA4* as a novel gene associated with IS. To confirm the pathogenicity of this gene, we used a zebrafish model and found that impaired EPHA4 pathway components and resulting defects in central pattern generators (CPGs) were associated with asymmetric spinal locomotion, which may serve as a contributory factor in the development of IS. Additionally, we searched for candidate genes related to EPHA4 signaling pathways and identified mutations in the *NGEF* among IS patients. Overall, our data suggest that the impairment of the EPHA4 pathway and CPGs plays a previously unknown role in the development and progression of IS.

## Results

### Enrichment analysis of rare variants in IS candidate genes

To identify candidate genes associated with IS, we conducted a comprehensive literature review of SNPs that are linked to IS from 14 published GWASs. This led us to obtain 41 SNPs that showed genome-wide significance ($p < 5 \times 10^{-8}$) as detailed in *Supplementary file 1*. We then employed positional mapping, expression quantitative trait locus (eQTL) mapping, or chromatin interaction mapping to identify 156 candidate genes that could be potentially linked to these SNPs. Further analysis of rare variants for the 156 candidate genes using exome data from 411 Chinese probands and 3800 unrelated Chinese controls identified *EPHA4* as the only significant gene (p=0.045, OR = 4.09) (*Supplementary file 2*). EPHA4 is a member of the Eph family of receptor tyrosine kinases which play a vital role in the development of the nervous system (*Kania and Klein, 2016*). We identified three rare variants in *EPHA4* from three IS patients, including one splicing-donor variant and two missense variants (*Supplementary file 2*). The flowchart of the entire pipeline is shown in *Figure 1—figure supplement 1*.

### Inheritance pattern and functional analyses of variants in *EPHA4*

We characterized the inheritance pattern of variants in *EPHA4* using either trio sequencing data or Sanger sequencing for the parents. None of the parents being tested had scoliosis based on a clinical screening test. Of the three rare variants in *EPHA4*, two (NM_004438.3: c.1443+1G>C and c.2546G>A [p.Cys849Tyr]) arose de novo (*Figure 1A* and *Table 1*). The heterozygous splicing variant (c.1443+1G>C) was identified in a 15-year-old female IS patient (SCO2003P0846) (*Figure 1B*). Scoliosis in this patient was diagnosed at 13 years of age, with a main curve Cobb angle of 60°. The in vitro minigene assay showed that the c.1443+1G>C variant introduced a new splicing site, resulting in a 36 bp in-frame deletion in exon 6 (*Figure 1G and I*, *Figure 1—figure supplement 2A*). The *EPHA4* heterozygous missense variant (c.2546G>A, p.Cys849Tyr), identified in a 13-year-old female IS patient (SCO2003P2146) (*Figure 1C*), is located in the protein tyrosine kinase domain of EPHA4 protein (*Figure 1A*). The onset of scoliosis in this patient was at 11 years of age with a 70° main curve Cobb angle. Given the critical role of EPHA4 in phosphorylating CDK5, which in turn activates downstream signaling pathways (*Fu et al., 2007*), a western blot analysis of the *EPHA4*-c.2546G>A variant was conducted to investigate the protein expression levels of EPHA4 and CDK5 and the amount of phosphorylated CDK5 (pCDK5) in HEK293T cells transfected with *EPHA4*-mutant or *EPHA4*-WT plasmid. Our results revealed that the missense variant resulted in a decreased phosphorylation level of CDK5 (p=0.015), suggesting a partial loss-of-function (LoF) of *EPHA4* (*Figure 1F*).

We next investigated de novo noncoding variants of *EPHA4* via whole-genome sequencing (WGS) in 116 trio families with IS. A de novo heterozygous *EPHA4* intronic variant (c.1318+10344A>G) was identified in a 16-year-old female patient (SCO2003P2080) (*Figure 1D* and *Table 1*). She developed scoliosis at 11 years of age with a 70° main curve Cobb angle. This variant is predicted to affect the branch point of the fifth intron of *EPHA4* (*Figure 1—figure supplement 2B*). We performed nested PCR to show that this variant induced exon 5 skipping, resulting in a 339 bp in-frame deletion (*Figure 1H and J*).

Next, we employed an in-house gene matching approach under the framework of the DISCO study, which identified a 6-year-old patient (SCO2003P3202) who had been previously diagnosed with Waardenburg syndrome caused by a 4.46 Mb de novo deletion at 2q35-q36.2 (*Li et al., 2015*;

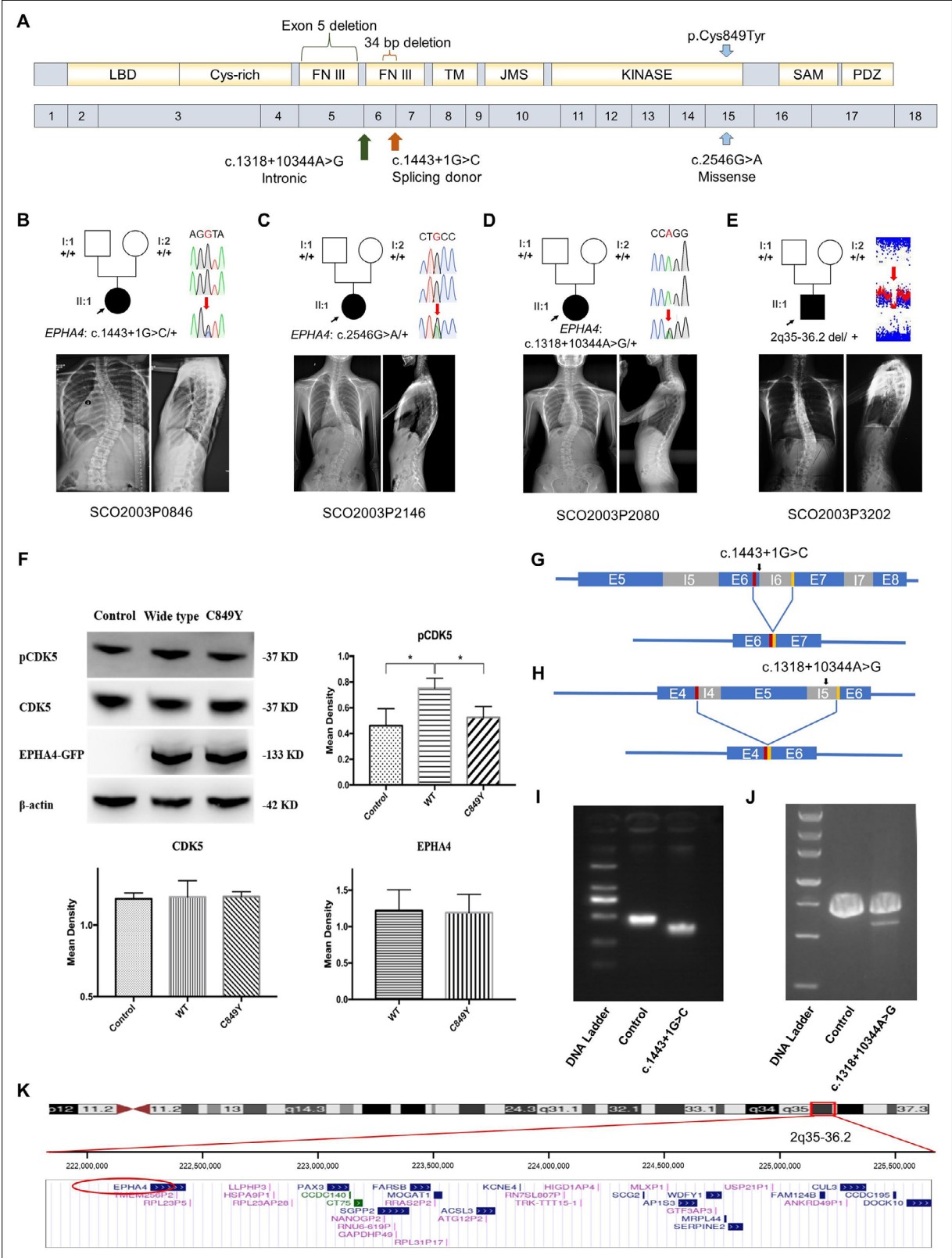

**Figure 1.** Clinical and genetic information on idiopathic scoliosis (IS) patients and functional effect of *EPHA4* variants. (**A**) Locations of three *EPHA4* single-nucleotide variants relative to the protein domains (top panel) and exons 1–18 (bottom panel). (**B-E**) Pedigrees and spinal radiographs of four probands with dominant gene variants. Sanger sequencing confirmed the variants. The arrows indicate the probands. The term +/+ denotes the wild-type, and cDNA change/+ denotes the heterozygous variant. (**F**) Western blot analysis of *EPHA4*-c.2546G>A variant showing the protein expression

*Figure 1 continued on next page*

*Figure 1 continued*

levels of EPHA4 and CDK5 and the amount of phosphorylated CDK5 (pCDK5) in HEK293T cells transfected with *EPHA4*-mutant or *EPHA4*-WT plasmid. WT: wild-type. Data represent three independent experiments. Error bars show mean ± SD. P<0.05 was considered statistically significant. (**G**) Schematic representation of the effect of the *EPHA4*-c.1443+1G>C mutation on the splicing process. This variant induced a new splicing site (red box). The yellow box indicates the splicing acceptor. (**H**) Schematic representation of the effect of the *EPHA4*-c.1318+10344A>G mutation on the splicing process. This variant induced a new splicing site (red box). The yellow box indicates the splicing acceptor. (**I**) The minigene assay result showed that the c.1443+1G>C variant introduced a new splicing site, resulting in a 36 bp in-frame deletion in exon 6. (**J**) The nested PCR showed that the c.1318+10344A>G variant induced exon 5 skipping, resulting in a 339 bp in-frame deletion. (**K**) NCBI RefSeq genes included in 2q35-q36.2 from UCSC Genome Browser. *EPHA4* is shown by the red oval.

The online version of this article includes the following source data and figure supplement(s) for figure 1:

**Source data 1.** Original western blots for *Figure 1F*.

**Figure supplement 1.** Flowchart for identification of causative genes.

**Figure supplement 2.** The splicing analysis by the Alamut software.

*Table 1*). Intriguingly, the patient also presented mild scoliosis (*Figure 1E*). The deletion included the entire *PAX3* gene, which was responsible for the Waardenburg syndrome phenotype, and 36 neighboring genes, including *EPHA4* (*Figure 1K*). As scoliosis is not typically associated with Waardenburg syndrome caused solely by *PAX3* pathogenic variants (*Tassabehji et al., 1993*), we hypothesize that the deletion of *EPHA4* may be responsible for the IS phenotype in this patient.

Notably, the GWAS signal which we mapped to *EPHA4* (rs13398147) (*Zhu et al., 2015*) represents a significant eQTL in esophagus and colon tissues, with the T allele associated with decreased expression of *EPHA4*. In our East Asian GWAS cohort of 6449 adolescent IS patients and 158,068 controls, we identified another two eQTLs in *EPHA4* associated with decreased expression of *EPHA4* in brain tissue (*Supplementary file 3*). In the same GWAS cohort, common SNPs in *EPHA4*, after aggregation, also showed significant enrichment (p=0.023) in IS patients vs controls. Taken together, the convergence between rare and common variants of *EPHA4* that lead to LoF or hypomorphic effects highlights the pivotal role of *EPHA4* in the pathogenesis of IS.

## IS-like phenotypes in zebrafish *epha4* mutants

To investigate the role of *EPHA4* in scoliosis, we utilized zebrafish as a model system due to its versatile nature in modeling adolescent IS (*Bagnat and Gray, 2020*; *Boswell and Ciruna, 2017*; *Grimes et al., 2016*; *Xie et al., 2022*). Zebrafish have two homologs of *EPHA4*, *epha4a* and *epha4b*. Using CRISPR-Cas9, we established a stable *epha4a* zebrafish mutant line with a 63 bp deletion in exon 3, which introduced a stop codon and resulted in a truncated protein (*Figure 2A and B*). The homozygous *epha4a* mutant larvae had no apparent defects in either notochord or body axis development (*Figure 2—figure supplement 1A*). Consistent with previous reports, the hindbrain rhombomeric boundaries were found to be defective in both the *epha4a* homozygous mutants and morphants (*Figure 2—figure supplement 1B*; *Cayuso et al., 2019*; *Cooke et al., 2005*). Interestingly, more than 75% adult mutants showed mild scoliosis (88 of 116), and some mutants exhibited severe scoliotic

**Table 1.** Dominantly inherited variants identified in *EPHA4* and *NGEF*.

Abbreviations: Chr: chromosomal localization; AA: amino acid; AD: autosomal dominant; CNV: copy number variant; ExAC PLI: probability of being loss-of-function intolerant from Exome Aggregation Consortium; CADD: Combined Annotation Dependent Depletion score; gnomAD: Genome Aggregation Database; NA: not applicable.

| Patient ID | Chr | Gene | Ethnicity | Inheritance pattern | cDNA change | AA change | Variant type | ExAC PLI | CADD | GnomAD frequency | In-house frequency |
|---|---|---|---|---|---|---|---|---|---|---|---|
| SCO2003P0846 | 2 | *EPHA4* | Chinese | De novo | c.1443+1G>C | NA | Splice donor | 1 | 21.4 | 0 | 0 |
| SCO2003P2146 | 2 | *EPHA4* | Chinese | De novo | c.2546G>A | p.Cys849Tyr | Missense | 1 | 28.2 | 0 | 0 |
| SCO2003P2080 | 2 | *EPHA4* | Chinese | De novo | c.1318+10344A>G | NA | Intronic | 1 | NA | 0 | 0 |
| SCO2003P3202 | 2 | *EPHA4* | Chinese | De novo | 2q35-36.2 4.6Mb deletion | NA | CNV | NA | NA | 0 | 0 |
| SCO2003P3332 | 2 | *NGEF* | Chinese | De novo | c.1A>G | p.Met1? | Start lost | 0.95 | 12.8 | 0 | 0 |
| TSRHC01 | 2 | *NGEF* | Non-Hispanic White | AD | c.857C>T | p.Ala286Val | Missense | 0.95 | 29.6 | 0.00002 | 0 |

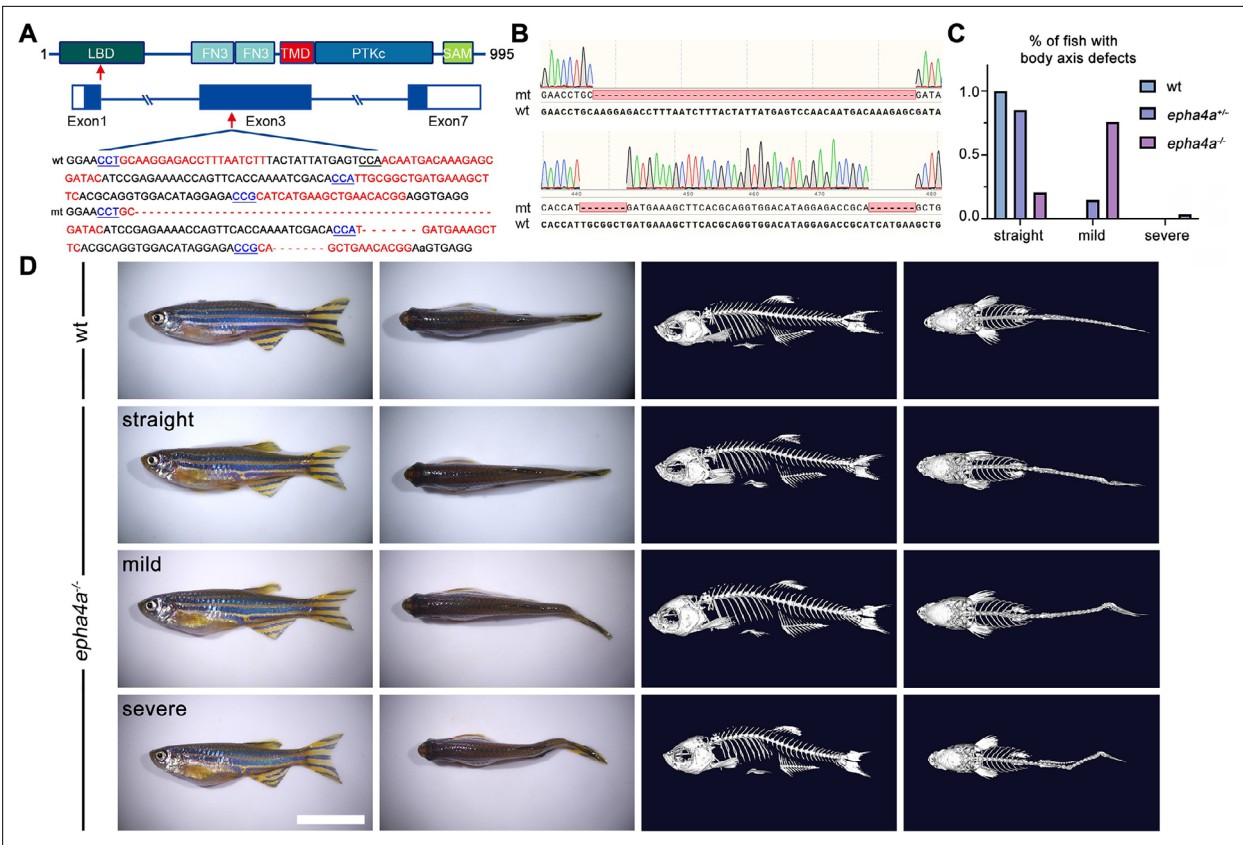

**Figure 2.** Scoliosis in zebrafish *epha4a* mutants. (**A**) Diagram of the protein domains, genomic structures, and sequences of wild-type and corresponding *epha4a* mutants. Red arrows indicate mutation sites. Blue boxes indicate open reading frames. Underlined sequences indicate the protospacer adjacent motif (PAM) region, and red fonts indicate Cas9 binding sites. LBD: ligand binding domain; FN3: fibronectin type 3 domain; TMD: transmembrane domain; PTKc: catalytic domain of the protein tyrosine kinases; SAM: sterile alpha motif. (**B**) Sanger sequencing results confirmed the deletion of the target region in *epha4a* mutant transcripts. (**C**) Bar graph showing the percentages of adult zebrafish with normal, mild, or severe body axis defects in wild-type (n=76), *epha4a* heterozygote (n=95), or *epha4a* homozygous (n=116). (**D**) Representative images of wild-type and homozygous *epha4a* mutants. Micro CT images are shown on the right. Lateral and dorsal views are shown. Scale bar: 1 cm.

The online version of this article includes the following figure supplement(s) for figure 2:

**Figure supplement 1.** Phenotypes of *epha4a* mutants.

**Figure supplement 2.** Zebrafish *epha4b* mutants exhibited body axis defects during development.

phenotype (4 of 116) (*Figure 2C and D*, *Videos 1 and 2*). Remarkably, some heterozygous adult mutants also developed mild scoliosis (14 of 95), whereas none of the wild-type fish showed any signs of scoliosis (0 of 76) (*Figure 2C and D*). Similarly, we further generated the *epha4b* mutants with a 25 bp deletion in exon 3, which resulted in a frameshift mutation (*Figure 2—figure supplement 2A*).

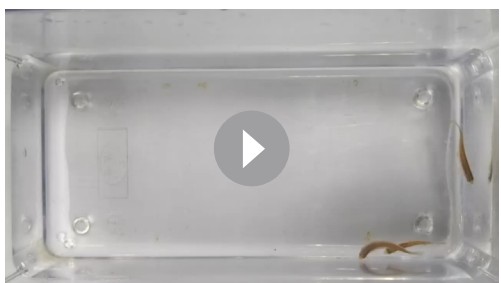

**Video 1.** Video showing the swimming of three wild-type fish.

https://elifesciences.org/articles/95324/figures#video1

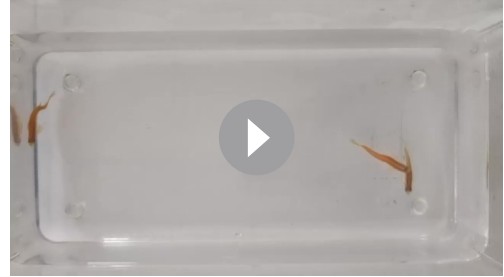

**Video 2.** Video showing the swimming of three epha4a mutants with severe or mild scoliosis.

https://elifesciences.org/articles/95324/figures#video2

The *epha4b* zebrafish mutants also developed mild scoliosis (28 of 43) (**Figure 2—figure supplement 2B and C**). Intriguingly, both *epha4a* and *epha4b* mutants exhibited early onset scoliosis starting from around 20 days post-fertilization (**Figure 2—figure supplement 2D**), a stage similar to that of IS patients. Collectively, these data suggest that mutations in Epha4 proteins are linked to the scoliotic phenotype in zebrafish.

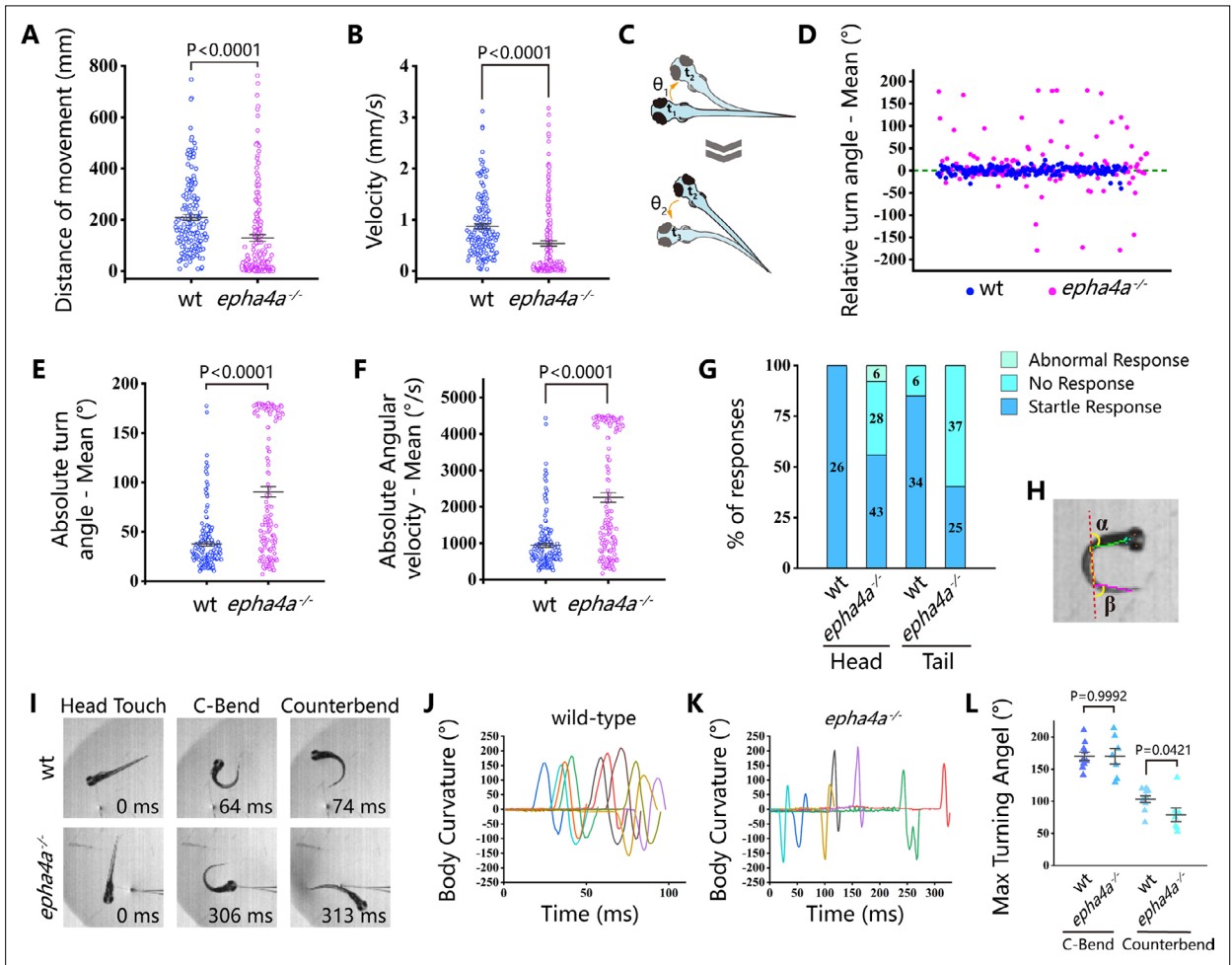

**Figure 3.** Abnormal left-right swimming pattern in the absence of Epha4a. (**A**) Dot plots showing the swimming distance of each 8 days post-fertilization (dpf) larva at a duration of 4 min (N=54 for wild-type and N=60 for *epha4a* mutants). (**B**) Dot plots showing the swimming velocity of wild-type and mutant larvae as indicated. (**C**) Diagram showing the turning angle ( $\theta$ ) of the larvae during swimming. (**D**) Scatter plot showing the relative turning angle of wild-type and mutant larvae. The relative angles were calculated by the sum of turning angles during fish swimming with left (positive) or right (negative) turns. The *epha4a* mutants favored turning to one side of their directions compared with those of wild-type larvae. (**E**) Dot plots showing the average absolute turning angle of wild-type and mutant larvae as indicated. (**F**) Dot plots showing the average absolute angular velocity of wild-type and mutant larvae as indicated. (**G**) Bar graph showing the percentages of 5 dpf zebrafish larvae with different reactions after tactile stimulation. N=10 for each group; the numbers of tactile stimulations are indicated in each column. (**H**) Representative images of the total body curvature measurements in zebrafish larvae, with values as the sum of α and β angles shown in the figure. (**I**) Representative time-series images of 5 dpf wild-type and *epha4a* mutant zebrafish larvae after tactile stimulation to the head. Each panel represents the points of maximal body curvature for the C-bend and counterbend after the tactile startle response. (**J, K**) A plot of body curvature angles as measured in panel (**H**) during swimming in response to tactile head stimulation in 5 dpf wild-type (panel J, N=5 larvae, n=10 stimuli) and *epha4a* mutants (panel K, N=5 larvae, n=7 stimuli). Each colored curve represents an independent experiment showing the response of a single larva to a stimulus. The positive angle means turning right. (**L**) The maximum curvature angles during the first C-bend and counterbend after tactile stimulation in wild-type and *epha4a* mutant larvae.

The online version of this article includes the following figure supplement(s) for figure 3:

**Figure supplement 1.** Rescue of the *epha4a* homozygous mutant phenotype by *epha4a* mRNA injection.

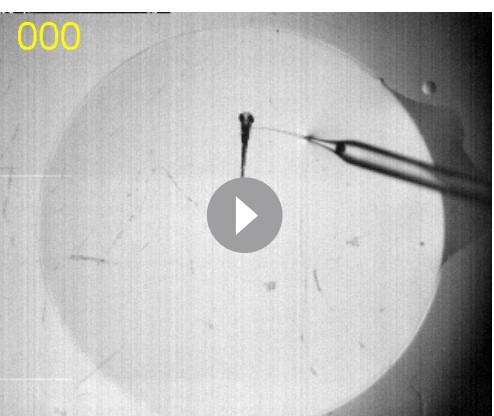

**Video 3.** High-speed video showing the startle response in a wild-type larva at 5 days post-fertilization (dpf) triggered by head tactile stimulation. Time units: ms.

https://elifesciences.org/articles/95324/figures#video3

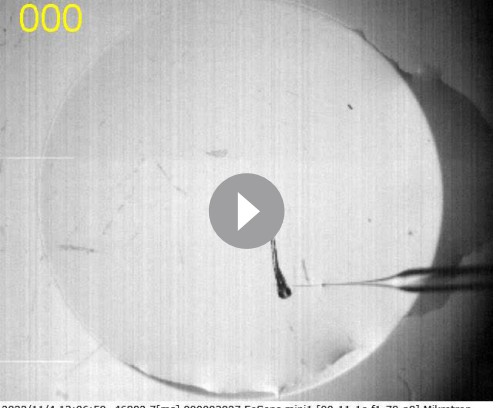

2022/11/4 13:06:50 −46882.7[ms] 000003927 EoSens mini1 [00-11-1c-f1-78-a8] Mikrotron 640x528(Q) 1000fps 326μs V1.4.12

**Video 4.** High-speed video showing the abnormal startle response of an *epha4a* mutant larva at 5 days post-fertilization (dpf) triggered by head tactile stimulation. Time units: ms.

https://elifesciences.org/articles/95324/figures#video4

## Abnormal left-right swimming pattern in the absence of Epha4a

While the *epha4a* mutants seemed to be grossly normal during the larvae stages, they developed spinal curvature gradually during later development. We decided to investigate whether these mutants exhibited any abnormalities in their behavior at the larvae stages using EthoVision XT software. By monitoring the swimming behavior of 8 days post-fertilization (dpf) zebrafish larvae, we found the motion distance and swimming velocity were significantly decreased in *epha4a* mutants (*Figure 3A and B*). In addition, we observed a remarkable difference in the relative turning angle and angular velocity between these two groups. The wild-type group changed their swimming direction randomly, showing a relative angle around 0° (*Figure 3C and D*). In contrast, the *epha4a* mutants favored turning to one side of their directions (a positive angle reflects a leftward turn) (*Figure 3C and D*). Moreover, the absolute turn angles and turning speed (angular velocity) were significantly higher in these mutants (*Figure 3E and F*). To rule out potential off-target effects, we injected *epha4a* mRNA into *epha4a* mutants, which significantly restored the swimming coordination defects (*Figure 3—figure supplement 1*).

Next, we compared the swimming behavior after startle response between wild-type and *epha4a* mutants. We used a needle to touch the head or the tail of 5 dpf zebrafish larvae to stimulate swimming behavior. In sibling controls, the larvae responded to the tactile stimulation and swam away quickly (*Video 3*). Conversely, the *epha4a* mutants failed to respond to the initial stimulation, and the swimming pattern was defective with an abnormal bending pattern (37 of 62 from tail-stimulation group and 34 of 77 from head-stimulation group) (*Figure 3G*, *Video 4*). We further performed a more comprehensive analysis by high-speed video microscopy. After tactile stimulation, wild-type larvae displayed a high-speed C-bend turn followed by a weaker counterbend turn after several milliseconds (*Figure 3H and I*). This rhythmic left/right swimming pattern ensures that the fish swim away from the frightening stimulus. The turning angles of the control larvae after stimulation changed with a sinusoidal wave pattern (*Figure 3J*). In contrast, this pattern was dramatically different in *epha4a* mutants (*Figure 3I and K*). Of note, although the C-bend turning angles were similar between *epha4a* mutants and control siblings, the turning angles of the counterbend decreased significantly (*Figure 3L*), implying a left/right coordination defect. Taken together, both regular swimming and tactile stimulation analyses suggested that the left-right coordination swimming pattern is compromised in the absence of Epha4a.

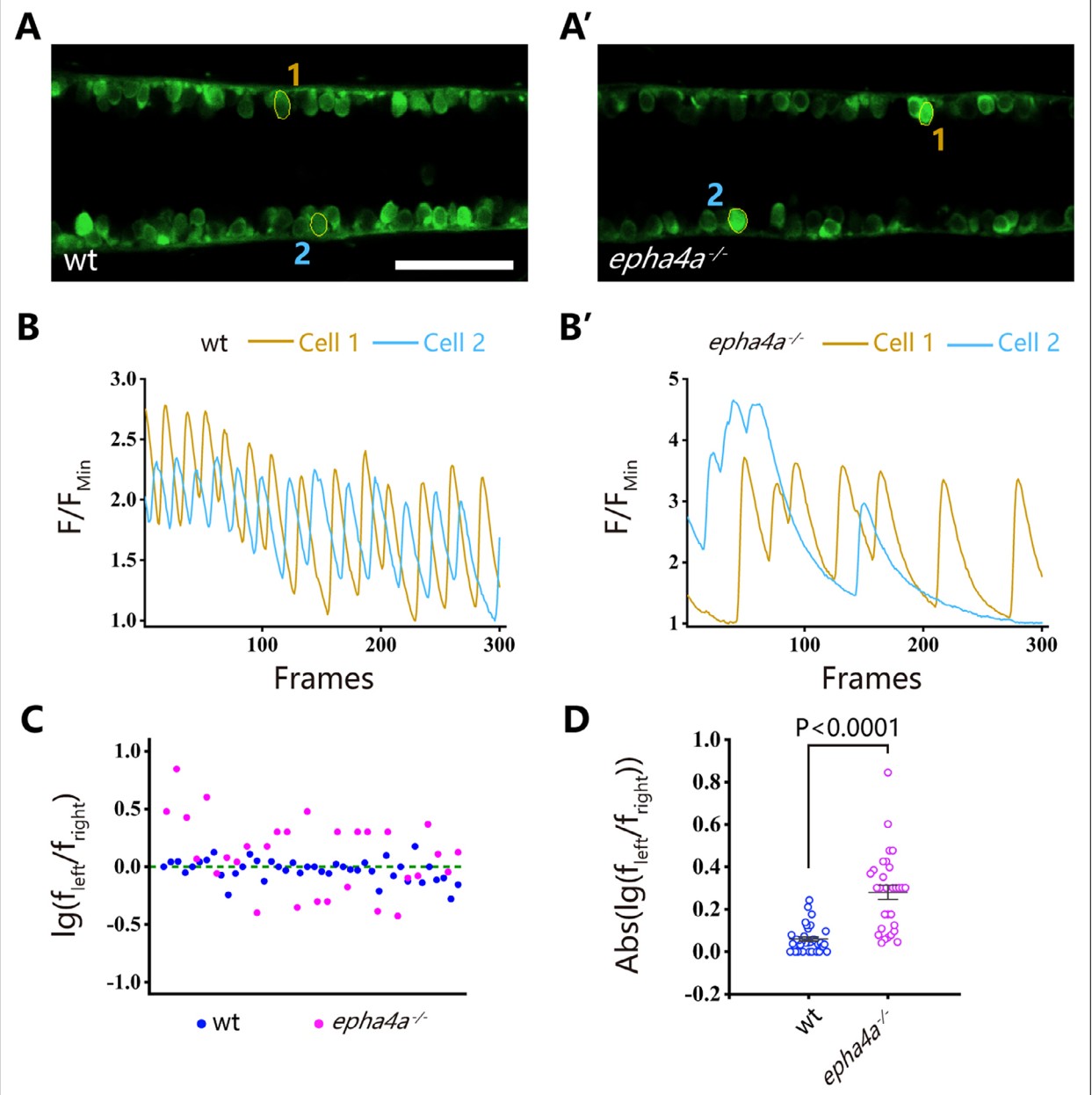

**Figure 4.** Uncoordinated left-right activation of spinal cord neurons in *epha4a* mutants. (**A**, **A'**) Fluorescent images showing the dorsal view of 24 hr post-fertilization (hpf) Tg(*elavl3*:GAL4; *UAS*:GCaMP) double transgenic larvae. The corresponding movies are shown in *Videos 5 and 6*. (**B**, **B'**) Line charts showing the quantification of fluorescence changes of the region of interests (ROIs, circled in **A**, **A'**) in wild-type larvae and *epha4a* mutants. (**C**) Scatter plot showing the distribution trend of the ratio of the calcium signal frequency between left and right in wild-type (N=14 larvae, n=42 experiments) and *epha4a* mutants (N=10 larvae, n=30 experiments). (**D**) Statistical graph of the ratio of the calcium signal frequency between left and right in wild-type larvae and *epha4a* mutants. Scale bars: 50 µm in panels (**A, A'**).

The online version of this article includes the following figure supplement(s) for figure 4:

**Figure supplement 1.** Uncoordinated left-right activation of spinal cord neurons in the absence of Epha4a.

## Defects of left-right coordination due to abnormal CPG in the absence of Epha4a

The coordinated left-right locomotion of zebrafish larvae relies on the synchronized contraction of muscle fibers, a process regulated by motor neurons situated on each side of the fish. To explore this intricate mechanism, we utilized a Tg(*elavl3*:GAL4; *UAS*: GCaMP) double transgene, allowing the expression of a genetically encoded calcium sensor in all neurons. We observed the rhythmic activation

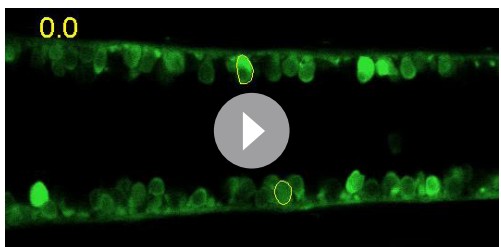

**Video 5.** Alternated activation of calcium signaling in motor neurons of 24 hr post-fertilization (hpf) wild-type Tg(*elavl3*:GAL4; *UAS*: GCaMP) transgenic larva.

https://elifesciences.org/articles/95324/figures#video5

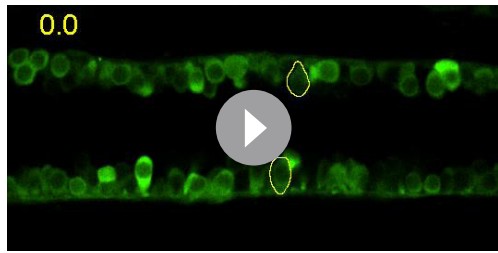

**Video 6.** Alternated activation of calcium signaling in motor neurons of 24 hr post-fertilization (hpf) wild-type Tg(*elavl3*:GAL4; *UAS*: GCaMP) transgenic larva injected with control MO.

https://elifesciences.org/articles/95324/figures#video6

of calcium signaling in motor neurons located within the spinal cord (*Figure 4A*, *Figure 4—figure supplement 1A*, *Videos 5 and 6*). In wild-type larvae, the calcium signals exhibited an alternating pattern between the left and right sides of the body, whereas this coordinated pattern was disrupted in both *epha4a* mutants and morphants (*Figure 4A–B vs A'–B'*, *Figure 4—figure supplement 1A–B vs A'–B'*, *Videos 7 and 8*). Additionally, we found that the activation frequency of motor neurons on the left and right sides was comparable in wild-type larvae, but significantly different in the absence of Epha4a (*Figure 4C and D*, *Figure 4—figure supplement 1C and D*).

One well-established concept regarding left-right coordination involves the presence of CPGs, which are regulated by interneuron circuitry within the spinal cord. We further examined the axon guidance of interneurons in *epha4a* mutants. First, we investigated the commissural trajectories of reticulospinal (RS) interneurons. In control larvae, the large Mauthner neurons, along with other RS neurons, were symmetrically positioned on each side of the midline and projected their axons to the contralateral sides (*Figure 5A*). These bilaterally projected axons typically crossed at the midline and subsequently synapsed on motoneurons of the opposite sides, contributing to the generation of spinal cord neural circuits (*Hale et al., 2016*). However, in *epha4a* mutant larvae, we observed an abnormal pattern in axonal projections. Specifically, the mutant axons failed to traverse the midline and instead extended ipsilaterally (*Figure 5A*). In addition, the distance between Mauthner neurons (rhombomere 4) and rhombomere 7 was significantly decreased, and rhombomere 5 was scarcely visible in the mutant larvae (*Figure 5A and B*). Furthermore, the sites of axon crossing between two Mauthner neurons tended to deviate to one side of the midline in the mutants (*Figure 5C–D'*).

Cerebrospinal fluid-contacting neurons (CSF-cNs) represent a unique type of interneuron responsible for modulating the V0 and V2a interneurons, which are integral components of locomotor CPGs (*Fidelin et al., 2015*; *Talpalar et al., 2013*; *Wu et al., 2021*). In wild-type larvae, we observed that the ascending axons of these neurons projected either to the right or left side from the midline, as visualized using Tg(*pkd2l1*:GAL4;*UAS*:Kaede) double transgenic larvae (*Figure 6A*). However, in the absence of Epha4a, the projection of these neurons exhibited notable disorganization, with numerous axons crossing the midline from one side of the trunk. This disorganized pattern was observed in both

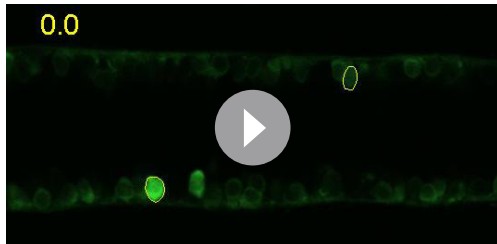

**Video 7.** Abnormal activation of calcium signaling in motor neurons of 24 hr post-fertilization (hpf) Tg(*elavl3*:GAL4; *UAS*: GCaMP) transgenic larva with homozygous *epha4a* mutant background.

https://elifesciences.org/articles/95324/figures#video7

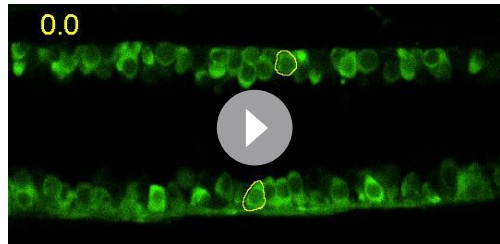

**Video 8.** Abnormal activation of calcium signaling in motor neurons of 24 hr post-fertilization (hpf) Tg(*elavl3*:GAL4; *UAS*: GCaMP) transgenic larva injected with *epha4a* MO.

https://elifesciences.org/articles/95324/figures#video8

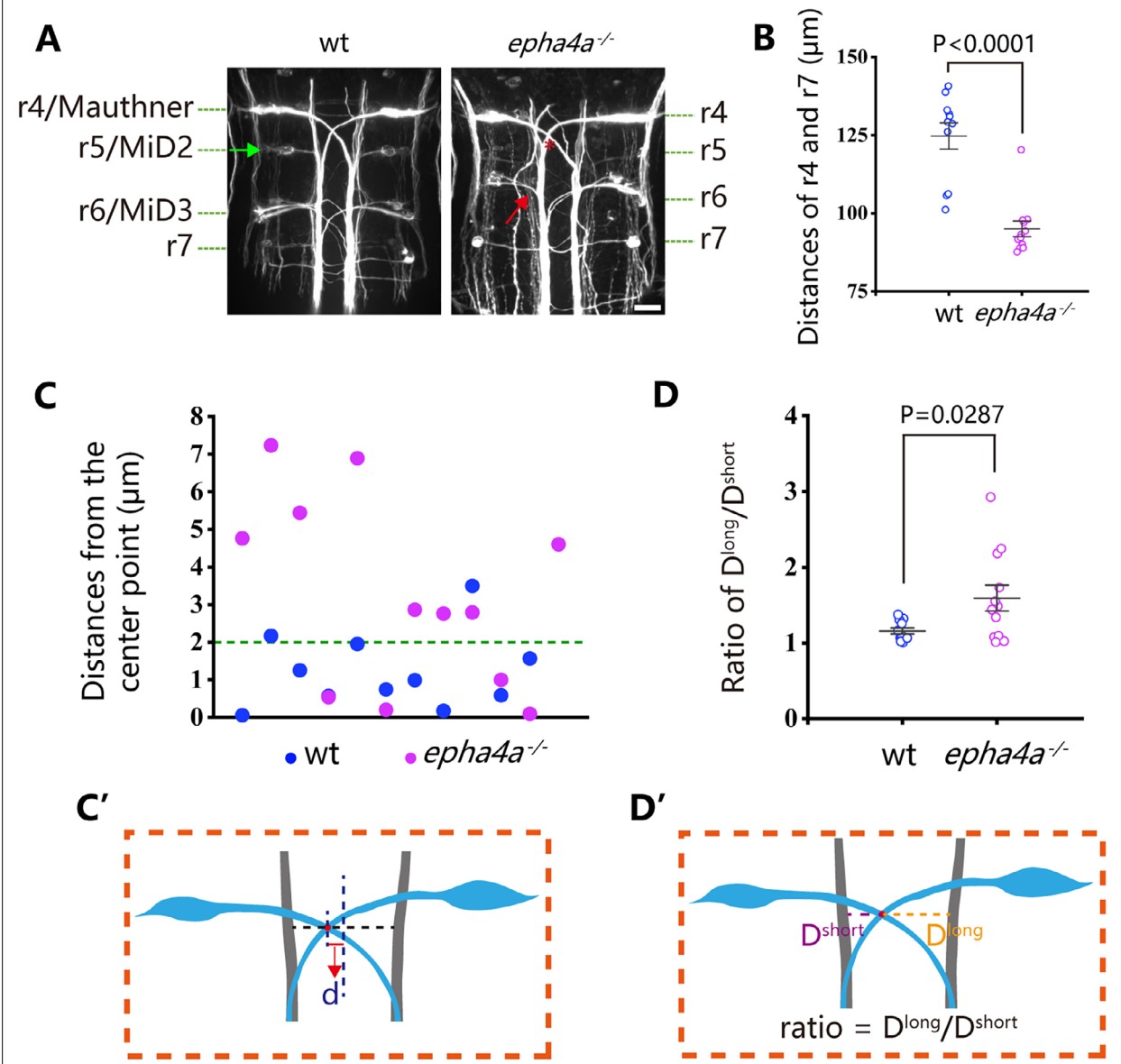

**Figure 5.** Disorganized neural patterning in *epha4a* mutants. (**A**) Confocal images showing reticulospinal neuronal axons in 48 hr post-fertilization (hpf) wild-type and *epha4a* mutant larvae visualized with anti-neurofilament antibody RMO44. Asterisks indicate the cross sites of Mauthner axons. The green arrow indicates the cell body of the r5/MiD2 neuron in a wild-type larva. The red arrow points to the ipsilaterally projected axon of r6/MiD3 in the mutant larva, which is normally projected to the other side in wild-type fish. (**B**) Statistical chart showing the distance between r4 and r7 of 48 hpf wild-type and *epha4a* mutants. (**C**) Scatter plot showing the distance (d) between the center line and the intersection site of Mauthner axons as indicated in panel **C'**. (**D**) The ratio of the distance between the intersection site of Mauthner axons and bilateral axon bundles in 48 hpf wild-type (N=11 larvae) and *epha4a* mutants (N=12 larvae). The ratios were calculated as in panel **D'**. Scale bars: 20 μm in panel (**A**).

heterozygotic and homozygotic mutants (*Figure 6A and B*). Additionally, we employed an optogenetic approach to activate these CSF-cNs, utilizing the Tg(Gal4$^{s1020t}$; *UAS*:ChR2) double transgene. Following optical stimulation, we observed robust tail oscillations as previously described (*Wyart et al., 2009*). In wild-type larvae, these tail oscillations exhibited a symmetrical left-right beating pattern (*Figure 6C and D*, *Video 9*). However, a striking disruption of this symmetry was observed in *epha4a* morphants, as they consistently beat toward one side of the trunk following optical stimulation (a positive angle reflects a rightward turn; *Figure 6C'–D' and E*, *Video 10*). Collectively, these findings demonstrate that the left-right coordination deficiencies observed in *epha4a* mutants arise from abnormal neural circuit formation, which consequently disrupts the integrity of the CPGs.

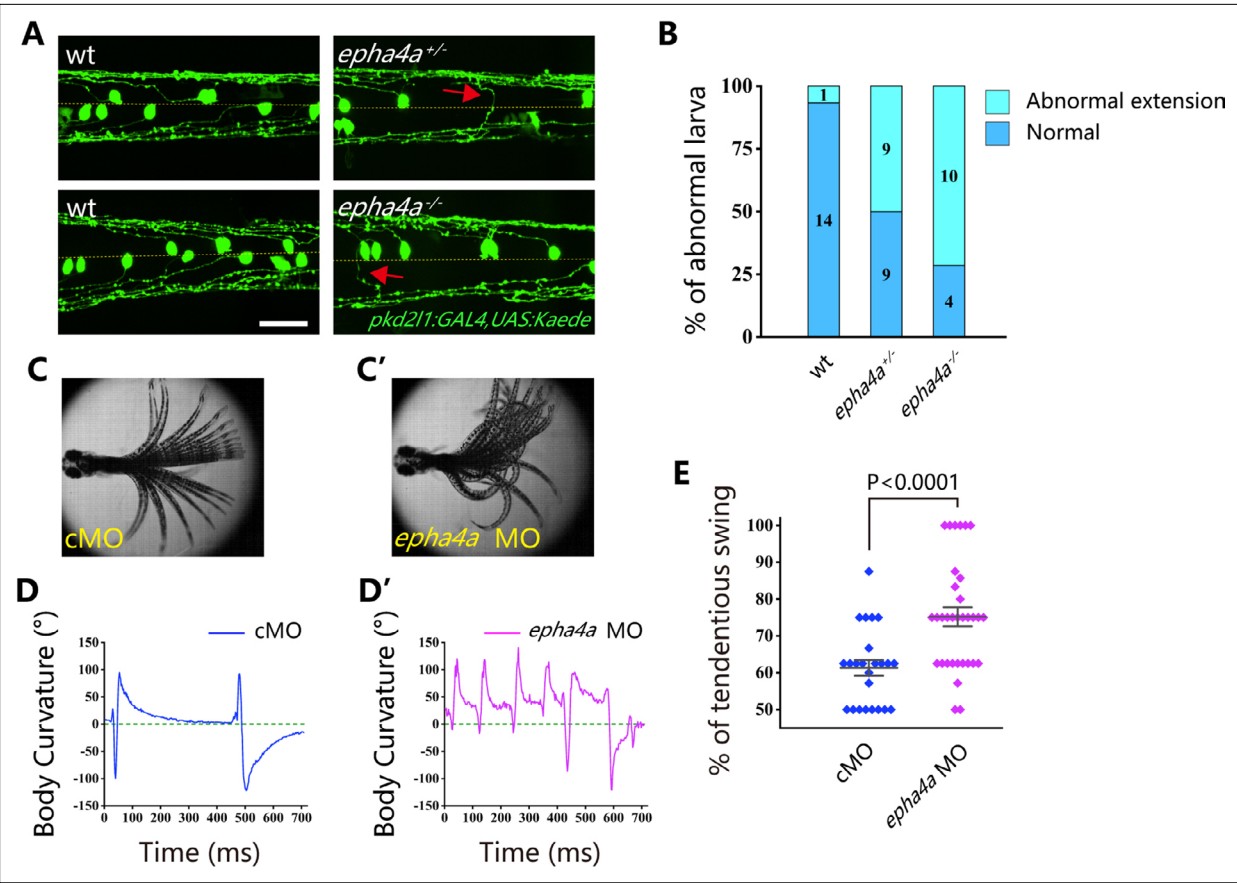

**Figure 6.** Aberrant swimming as a result of abnormal extension of cerebrospinal fluid-contacting neurons (CSF-cNs) axons. (**A**) Fluorescent images showing the distribution of ascending axons of CSF-cNs marked by Tg(pkd2l1:GAL4,UAS:Kaede) in 2 days post-fertilization (dpf) *epha4a* mutant larvae. Yellow line indicates the midline, and the red arrows indicate aberrantly extended axons in *epha4a*⁺/⁻ and *epha4a*⁻/⁻ larvae. (**B**) Bar graph showing the percentages of abnormal extension of CSF-cNs axons in 2 dpf wild-type, *epha4a* heterozygote, and *epha4a* homozygous. The numbers of larvae are indicated in each column. (**C**, **C'**) Superimposed frames of tail oscillations in 5 dpf control and *epha4a* morphants. (**D**, **D'**) A plot of body curvature angles in panels (**C**) and (**C'**). The positive angle means turning right. (**E**) Percentages of tendentious swing in control (N=8 larvae, n=24 experiments) and *epha4a* morphants (N=11 larvae, n=33 experiments). The percentages were calculated by the ratio of tendentious tail oscillation during the first eight swings. Scale bars: 50 μm in panel (**A**).

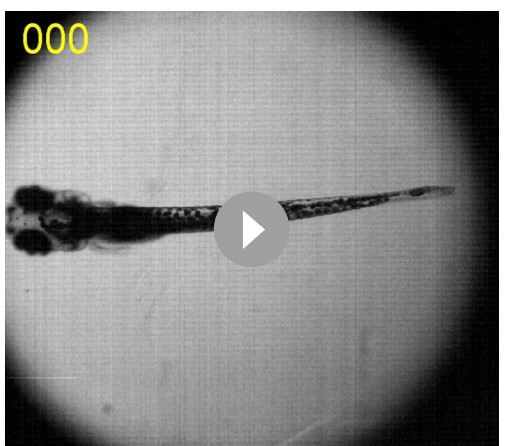

**Video 9.** Tail oscillation after light activation of 5 days post-fertilization (dpf) Tg(Gal4ˢ¹⁰²⁰ᵗ; UAS:ChR2) double transgene larva.
https://elifesciences.org/articles/95324/figures#video9

## Ephrin B3-Epha4 signaling regulates interneuron axon extension

To further explore the role of Epha4a during interneuron axon extension, we examined the expression of *epha4a* during early zebrafish embryonic development. Whole-mount in situ hybridization results showed that both *epha4a* and *epha4b* were abundantly expressed in the zebrafish spinal cord (*Figure 7—figure supplement 1A*). We plotted the expression of these two genes using two published single-cell transcriptome data, which showed that *epha4a* and *epha4b* were both expressed in interneurons, suggesting a role for *epha4* in interneuron function (*Figure 7—figure supplement 1B*; *Cavone et al., 2021*; *Scott et al., 2021*). Notably, the expression of *efnb3b*, encoding the ligand of Epha4, was highly enriched

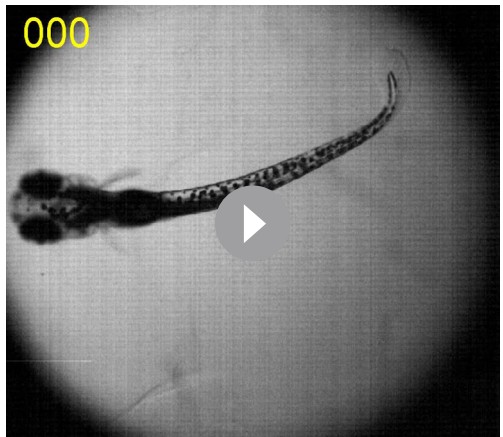

**Video 10.** Tail oscillation after light activation of 5 days post-fertilization (dpf) Tg(Gal4^s1020t; *UAS*:ChR2) double transgene larva injected with *epha4a* MO.

https://elifesciences.org/articles/95324/figures#video10

in the midline floor plate cells (*Figure 7—figure supplement 1B*).

Ephrins, through interacting with Eph receptors, play a critical role in repulsive axon guidance during neural development (*Egea and Klein, 2007*; *Flanagan and Vanderhaeghen, 1998*). We further analyzed axon guidance in *efnb3b* morphants. Similar to those of *epha4a* mutants, the *efnb3b* morphants also displayed axon guidance defects, as well as uncoordinated calcium activation (*Figure 7A–C*). In addition, morphants larvae also displayed left-right oscillation defects after optogenetic stimulation (a positive angle reflects a rightward turn) (*Figure 7D and E*).

## Candidate variants in the *EPHA4*-related genes

Our zebrafish studies suggested that EPHA4 signaling is crucial for interneuron axon guidance, hence the formation of functional CPGs. Next, we further asked whether mutation of other components of the EPHA4 signaling can result in IS in humans. By searching for rare variants in *EPHA4*-related genes (*Figure 8—figure supplement 1*; *Szklarczyk et al., 2019*), we identified a heterozygous de novo start-loss variant (c.1A>G, p.Met1?) in *NGEF* in a 17-year-old male (SCO2003P3332) (*Figure 8A and B*, *Table 1*), whose scoliosis was diagnosed at

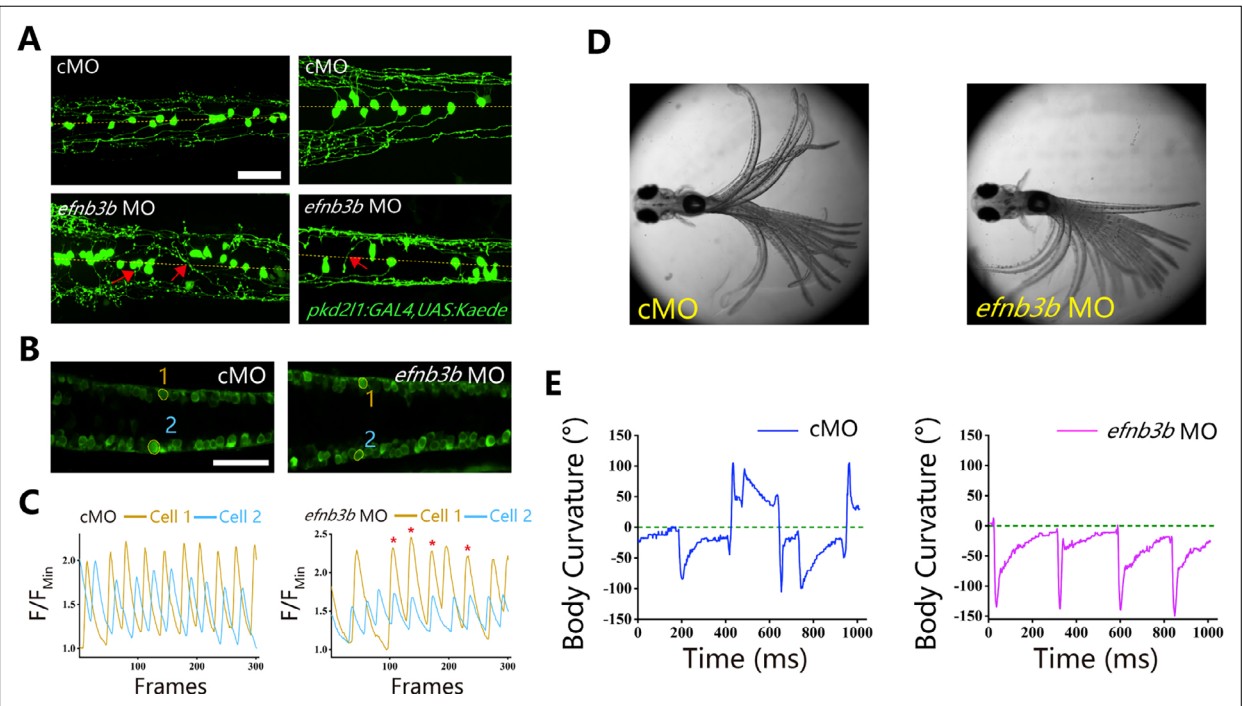

**Figure 7.** Left-right coordination defects in *efnb3b* morphants. (**A**) Fluorescent images showing the distribution of ascending axons of cerebrospinal fluid-contacting neurons (CSF-cNs) marked by Tg(*pkd2l1:GAL4,UAS:Kaede*) in 2 days post-fertilization (dpf) *efnb3b* morphants. The red arrows indicate aberrantly extended axons in *efnb3b* morphants. (**B**) Fluorescent images showing the dorsal view of 24 hr post-fertilization (hpf) Tg(*elavl3*:GAL4; *UAS*:GCaMP) transgenic larvae. (**C**) The line chart showing the quantification of fluorescence changes of the region of interests (ROIs) in control morphants and *efnb3b* morphants as indicated in panel **B**. (**D**) Superimposed frames of tail oscillations in 5 dpf control and *efnb3b* morphants. (**E**) A plot of body curvature angles in panel (**D**). The positive angle means turning right. Scale bars: 50 µm in panels (**A, B**).

The online version of this article includes the following figure supplement(s) for figure 7:

**Figure supplement 1.** Expression pattern of *epha4* and *efnb3* in the spinal cord.

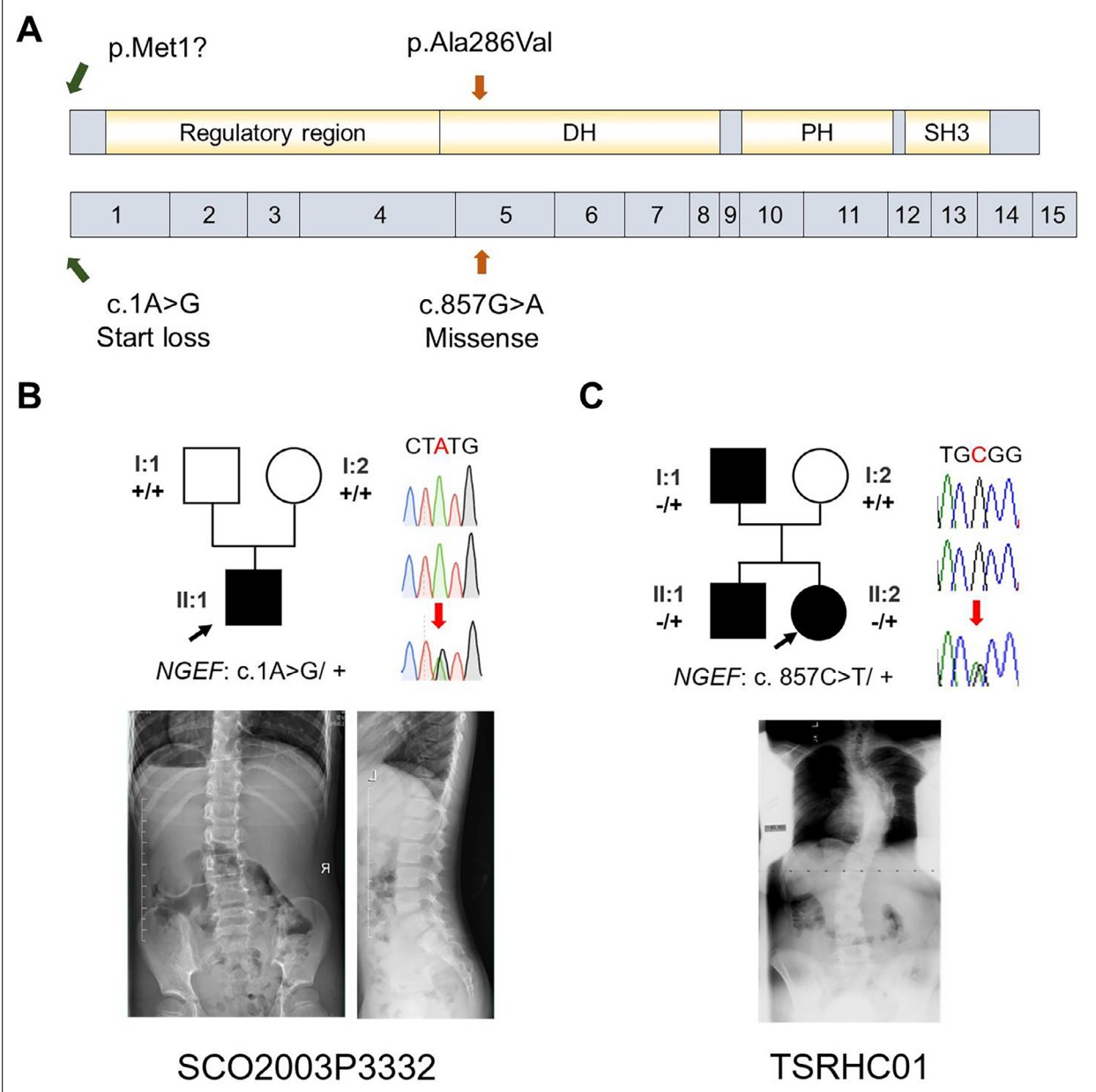

**Figure 8.** Idiopathic scoliosis (IS) patients with potential *NGEF* variants. (**A**) Protein structure of NGEF protein with the position of potential variants. (**B**, **C**) Pedigrees and spinal radiographs of two probands with dominant gene variants. Sanger sequencing results are shown on the right.

The online version of this article includes the following figure supplement(s) for figure 8:

**Figure supplement 1.** Proteins in STRING that interact with EPHA4.

15 years of age with a main curve Cobb angle of 60°. *NGEF* encodes the neuronal guanine nucleotide exchange factor Ephexin that differentially affects the activity of GTPases RHOA, RAC1, and CDC42. The activation of Ephexin is triggered by ephrin through EPHA4 (*Shamah et al., 2001*).

Strikingly, in a European-ancestry IS cohort, we further identified a dominant missense variant (c.857G>A, p.Ala286Val) in *NGEF* in a quad family with three affected members (*Figure 8C*, *Table 1*). The proband (II:2, TSRHC01) with onset of scoliosis at 14 years of age has a 58° main curve Cobb angle. This variant, which is located in the RhoGEF domain of Ephexin, is predicted to be highly deleterious (CADD = 29.6).

Altogether, our results suggest that defects of the CPGs owing to abnormal EPHA4 signaling maybe one of the crucial factors responsible for IS.

## Discussion

IS is a disease with diverse causes, and the underlying mechanisms can vary even among patients with similar scoliotic phenotypes. Previous genetic studies have highlighted the significance of the extracellular matrix (ECM) in maintaining the balance of axial bone and supporting soft tissues in the spine, thus playing a crucial role in IS development (*Haller et al., 2016*). For example, the top SNP associated with IS maps to *LBX1*, an essential molecule for ECM maintenance and bone homeostasis (*Takahashi et al., 2011*). Additionally, genetic loci in muscle development-related genes have also been associated with the onset of scoliosis, emphasizing the intricate interaction between bones and muscles (*Ogura et al., 2015*). However, it is worth noting that the variants in these genes explain only a small fraction of the overall heritability of IS. Consequently, it is imperative to establish connections between the extensive genetic findings and biological mechanisms that can elucidate the etiological landscape of IS.

In this study, we mapped 41 significant genome-wide loci to functional genes through positional mapping and functional mapping such as eQTL. Then, we determined the enrichment in patient cohorts of rare variants in these genes, which may have greater impacts compared with common SNPs. This approach revealed the convergence of SNPs and rare variants in *EPHA4* that are enriched in patients with IS. We also identified additional high-impact variants in *NGEF*, which is involved in the EPHA4 pathway.

Our studies using zebrafish have revealed that deficiency of Epha4 can lead to the development of scoliosis. Interestingly, we observed that even heterozygotic *epha4a* zebrafish mutants displayed mild scoliosis (*Figure 2C*), which is consistent with the occurrence of this condition in scoliosis patients. Further analysis involving behavior and imaging demonstrated that the absence of Epha4a resulted in defective left-right coordination. This coordination is crucially governed by CPGs, which generate rhythmic patterns of neural activity to coordinate limb movements on both sides of the body (*Kiehn, 2006*; *Marder and Bucher, 2001*; *Talpalar et al., 2013*). Previous studies have reported the involvement of ephrin and its receptors in axon guidance during the maturation of neural circuits, including CPGs (*Andersson et al., 2012*; *Borgius et al., 2014*; *Iwasato et al., 2007*).

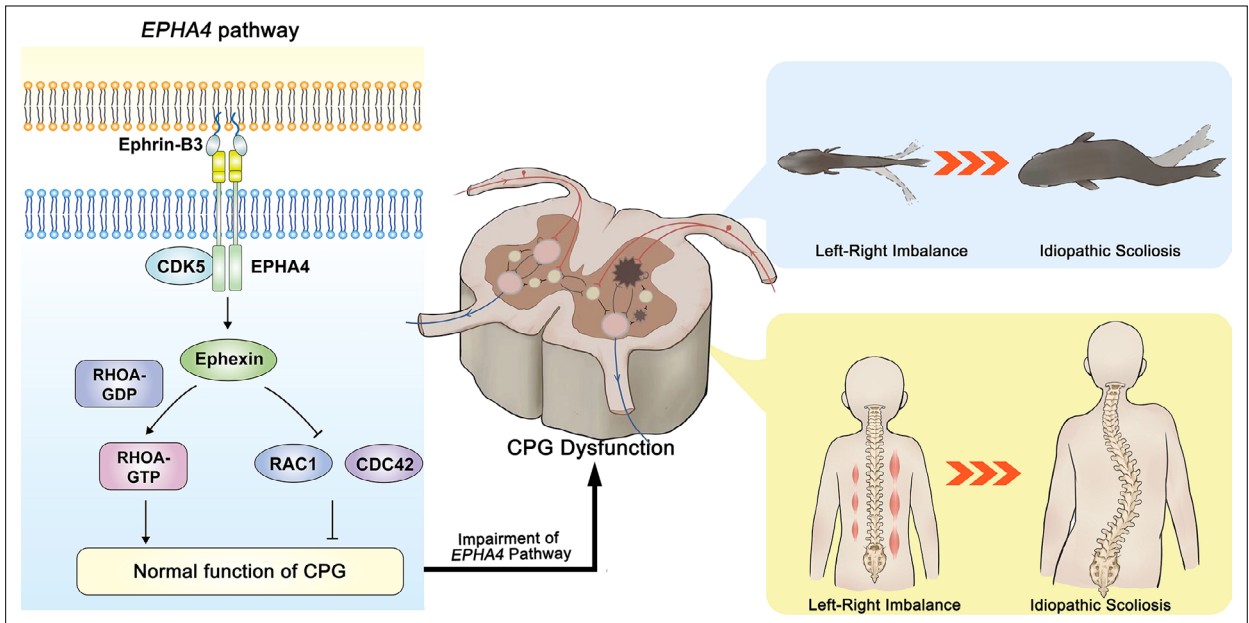

**Figure 9.** The proposed mechanism of idiopathic scoliosis (IS) mediated by *EPHA4* dysfunction. In healthy individuals, EphrinB3-activated EPHA4 phosphorylates CDK5, leading to the phosphorylation of Ephexin, a protein encoded by NGEF. Phosphorylated Ephexin can regulate axon guidance through either activating RHOA or suppressing CDC42 and RAC1 signaling. These processes are critical in maintaining the normal function of the central pattern generator (CPG), the local neural network that provides coordinated bilateral muscle control. Impairment of the EPHA4 pathway and CPG may cause an imbalance of the motor drive from the spinal cord during development, thus causing the uncoordinated left/right swimming behavior in zebrafish larvae and the asymmetry of the bilateral muscular pull in a young child. Although the appearance is normal in early childhood, the dysfunction produces a scoliotic curve during the growth spurt.

Ephrin B3, present in radial glial cells at the dorsal midline, serves as a repulsive barrier to axons expressing Epha4 (*Imondi et al., 2000*). Axon guidance is disrupted when this repulsive pathway is inactivated, leading to midline invasion (*Butt et al., 2005*; *Kullander et al., 2001*; *Paixão et al., 2013*). In *Epha4* and *Efnb3* knockout mice, a 'hopping' gait is observed, which is attributed to the aberrant crossing of motor neurons expressing Epha4 (*Kullander et al., 2003*). Consistent with these observations, we observed disrupted axon guidance in the interneurons, which are integral components of CPGs, in both *epha4a* mutants and *efnb3b* morphants. This finding suggests that CPG malfunction may be a significant contributing factor to the scoliosis phenotype observed in these genetic backgrounds. It is highly likely that the lack of coordinated left-right locomotion generates imbalanced mechanical forces on the spine, gradually leading to spinal curvature during later stages of development (*Figure 9*).

Thus, our data provided a novel biological mechanism of IS, i.e., the impairment of neural patterning and CPG. Previous studies have provided clues on the role of CPGs in IS. Patients with adolescent IS showed asymmetric trunk movement during gait, as characterized by increased relative forward rotation of the right upper body in relation to the pelvis (*Kramers-de Quervain et al., 2004*; *Nishida et al., 2017*). An electromyography (EMG) study also showed asymmetric activation of paraspinal muscles between the convex and concave sides at the scoliosis curve apex (*Shimode et al., 2003*). In a child with a strong family history of IS, asymmetric hyperactivity was observed by EMG months before scoliosis was evident (*Valentino et al., 1985*). These left-right locomotor coordination abnormalities indicated the maldevelopment of CPGs as a potential cause of IS. The CPG asymmetry may induce an imbalance in trunk muscle strength, resulting in asymmetric rib drooping. This leads to an abnormal vertebral rotation and then the onset of IS. Moreover, in a companion study, Wang et al. identified a number of rare variants in *SLC6A9*, which encodes glycine transporter 1 (GLYT1), in familial and sporadic adolescent IS cases. The *slc6a9* mutant zebrafish also exhibited discoordination of spinal neural activities with pronounced lateral spinal curvature, recapitulating the human IS phenotype (*Wang et al., 2024*). Taken together, we propose that the dysfunction of CPGs would cause an imbalance in the motor drive from the spinal cord and the asymmetric transversospinalis muscle pull, eventually producing a scoliotic curve (*Figure 9*).

In summary, our study demonstrates that both common and rare variants within the EPHA4 pathway contribute to the genetic architecture of IS. EPHA4 pathway dysfunction causes axon pathfinding defects, resulting in impaired coordinated left-right locomotion by disrupting neural patterning and the function of CPGs, thereby potentially leading to IS.

## Materials and methods
### Mapping of candidate genes utilizing previous association studies

Through a systematic literature review, we identified IS-associated SNPs reported in GWASs and meta-analyses of GWAS. The literature search was carried out using MEDLINE (via https://pubmed.ncbi.nlm.nih.gov/) and Web of Science (via Clarivate Analytics) and was limited to English-language articles published from January 1980 to October 2020. The following keywords were combined to perform the search: 'idiopathic scoliosis' AND 'GWAS' OR 'SNP' OR 'single nucleotide polymorphism' OR 'variant' (*Supplementary file 4*). The inclusion/exclusion criteria of abstracts are provided in *Supplementary file 5*. After screening the titles and abstracts, we obtained from the full-text articles the rsID, chromosome, and position for SNPs with a threshold for genome-wide significance of $p<5.0 \times 10^{-8}$. For SNPs identified in multiple studies, we recorded the lowest p-value.

SNPs reported in previous studies were first pruned using Functional Mapping and Annotation (FUMA, v1.4.1, https://fuma.ctglab.nl/; *Watanabe et al., 2017*). Significant SNPs were considered independent at $r^2<0.6$. All known SNPs (available in the 1000 Genomes reference panel, https://www.internationalgenome.org/) were included for further gene mapping if they were in a linkage disequilibrium (LD) block ($r^2\geq0.6$) with a significant independent SNP. Three SNP-to-gene strategies were used:

1. For positional mapping of significant independent SNPs, we used annotations obtained from ANNOVAR (http://annovar.openbioinformatics.org). A 10 kb maximum distance was applied for intergenic SNPs.

2. For the eQTL mapping, we mapped significant independent SNPs and SNPs in an LD block to eQTLs across 44 GTEx tissue types (release v8) (*Aguet et al., 2020*). SNP-gene pairs with a false discovery rate (FDR)≤0.05 were considered significant.

3. For chromatin interaction mapping, we overlapped the significant independent SNPs and those in LD blocks with one end of significantly interacting regions across various tissue and cell types. Information on the interacting regions was derived from Hi-C datasets of 21 tissues and cell types provided by GSE87112. The significance of interactions for Hi-C datasets was computed by Fit-Hi-C, with an FDR≤$10^{-6}$ considered significant. Genes were mapped if their promoter regions overlapped with another end of the significant interactions. The promoter region was defined as the region from −250 bp to +500 bp relative to the transcription start site.

## Cohort description

The Peking Union Medical College Hospital (PUMCH) cohort: The PUMCH cohort comprised 411 unrelated Chinese patients with severe IS (Cobb angle≥40°) who underwent spinal surgery in the PUMCH between October 2017 and March 2022 as part of the Deciphering disorders Involving Scoliosis and COmorbidities (DISCO) study (http://www.discostudy.org/). The clinical diagnosis was confirmed using standing full-spine radiographs, three-dimensional computed tomography, and magnetic resonance imaging. These patients did not show any congenital or neuromuscular defect at the time of recruitment. The control cohort consisted of 3800 individuals without observable scoliosis and with exome sequencing or genome sequencing performed at PUMCH for clinical or research purposes. Individuals with vertebrae malformation or congenital developmental defects were excluded. A patient with a 2q35-36.2 deletion, including the *EPHA4* gene, was also recruited through in-house gene matching of the DISCO study (*Li et al., 2015*). This patient was subsequently evaluated for scoliosis by examination and radiography.

The East Asian cohort: The East Asian cohort includes totally 6449 IS patients and 158,068 controls from four independent datasets (Japanese dataset 1, *Kou et al., 2013*; *Takahashi et al., 2011*): 1261 cases, 15,019 controls; Japanese dataset 2 (*Ogura et al., 2015*): 878 cases, 21,334 controls; Japanese dataset 3 (*Kou et al., 2019*): 3333 cases, 119,630 controls; Hong Kong dataset: 977 cases, 2085 controls. The inclusion criteria for IS subjects were the same as our previous studies (*Fan et al., 2012*; *Kou et al., 2019*).

The Texas Scottish Rite Hospital for Children (TSRHC) cohort: We used a replication cohort of European-ancestry patients with IS from the TSRHC. Cases considered for inclusion in the study met criteria for a positive diagnosis of IS: lateral deviation from the midline greater than 15° as measured by the Cobb angle method from standing spinal radiographs, axial rotation toward the side of the deviation, and exclusion of relevant coexisting diagnoses.

## Blood sample collection

In the PUMCH cohort, genomic DNA samples were extracted from peripheral blood leukocytes of each subject using a QIAamp DNA Blood Mini Kit (QIAGEN, Hilden, Germany), according to the manufacturer's protocol. Purified DNA was qualified by NanoDrop 2000 (Thermo Fisher Scientific, Waltham, MA, USA) and quantified by Qubit 3.0 using the dsDNA HS Assay Kit (Life Technologies, Carlsbad, CA, USA). DNA samples were stored at 4°C until used.

In the East Asian and TSRHC cohort, genomic DNA was extracted from peripheral blood or saliva using a standard protocol.

## DNA sequencing and variant calling

In the PUMCH cohort, whole exome sequencing or WGS was performed on peripheral blood DNA from all individuals and available family members (*Supplementary file 6*). A SureSelect Human All Exon V6+UTR r2 core design (91 Mb, Agilent) was used for exon capture. The exomes were then sequenced on an Illumina HiSeq 4000 (Illumina, San Diego, CA, USA) according to the manufacturer's instructions. For WGS, sequencing libraries were prepared using the KAPA Hyper Prep kit (KAPA Biosystems, Kusatsu, Japan) with an optimized manufacturer's protocol. We performed multiplex sequencing using an Illumina HiSeq X-Ten sequencer (Illumina, San Diego, CA, USA). Variant calling and annotation were done in-house using the Peking Union Medical College Hospital Pipeline (PUMP) as described previously (*Chen et al., 2021*; *Zhao et al., 2021*).

In the TSRHC cohort, subjects in 162 adolescent IS families were sequenced as part of the Gabriella Miller Kids First Pediatric Research Consortium (GMKF) at The HudsonAlpha Institute for Biotechnology (Huntsville, AL, USA). In summary, DNA was normalized, sheared, and then ligated to Illumina paired-end adaptors. The purified ligated DNA was amplified, and exome sequencing was performed on the Illumina HiSeq X platform. The sample's sequences were aligned into GRCh38, and genotypes were joint-called per each family by the GMKF's Data Resource Center (DRC) at Children's Hospital of Philadelphia following GATK best practices as detailed here (*Mukhopadhyay et al., 2020*). GMKF DRC's alignment and joint genotyping pipelines are open source and made available to the public via GitHub (https://github.com/kids-first/kf-alignment-workflow copy archived at *Miller, 2025a*, and https://github.com/kids-first/kf-jointgenotyping-workflow copy archived at *Miller, 2025b*).

## Rare variant association analysis of the candidate genes

To determine the contribution of rare variants in GWAS candidate genes to IS, we analyzed the gene-based mutational burden for 156 candidate genes. Only ultra-rare variants with a gnomAD population-max allele frequency ≤0.01% and a cohort allele count ≤3 were analyzed. Each variant was assigned a weight, and the mutational burden of a given gene was defined as the maximum weight value among all ultra-rare variants carried by the individual. The SAIGE-GENE+ package was used to determine the weighted mutational burden test for each gene (*Zhou et al., 2022*).

Weighting criteria for the weighted burden analysis were developed according to the variant types and in silico results. The LoF variants (canonical splicing variants, nonsense variants, or variants that cause frameshift, stop-gain, or start-loss) were calculated together with the protein-altering variants, including missense variants and in-frame indels. Each variant was assigned a weight range from 0 to 1, and the mutational burden for a given gene was defined as the maximum weight value among all ultra-rare variants carried by the individual. The LoF variants annotated as 'high confidence' by the loss-of-function transcript effect estimator (LOFTEE) (*Karczewski et al., 2020*) were assigned a weight value of 1.0. The LoF variants annotated as 'low confidence' or unlabeled by LOFTEE, the noncanonical splicing variants with a SpliceAI score >0.5, and the missense variants with a rare exome variant ensemble learner (REVEL) (*Ioannidis et al., 2016*) score of 0.8 were assigned a weight value of 0.8. The in-frame insertions/deletions (indels) with a Combined Annotation Dependent Depletion (CADD) (*Kircher et al., 2014*) score >20 and the missense variants with a REVEL score >0.6 and ≤0.8 were assigned a weight value of 0.6. The in-frame indels with a CADD score >10 and ≤20 and the missense variants with a REVEL score >0.4 and ≤0.6 were assigned a weight value of 0.4. The in-frame indels and the missense variants with a REVEL score >0.2 and ≤0.4 were assigned a weight value of 0.2. The remaining missense variants were assigned a weight value of 0.

## Sanger sequencing of familial participants

Sanger sequencing of familial participants was performed to determine the origin of variants in *EPHA4* and *NGEF*. All LoF variants and protein-altering variants identified in familial participants were validated. Variant-encoding gene regions were amplified by PCR from genomic DNA obtained from probands, as well as from parents for trios, to determine the origin of the variants. The amplicons were purified using an Axygen AP-GX-50 kit (Corning, NY, USA) and sequenced by Sanger sequencing on an ABI 3730xl instrument (Thermo Fisher Scientific, Waltham, MA, USA).

## Minigene assay

The splicing variant (*EPHA4:* c.1443+1G>C) was characterized by a minigene assay. Genomic DNA from the heterozygous patient was amplified by PCR using a high-fidelity DNA polymerase. Amplicons included exon 6, intron 6, exon 7, intron 7, and exon 8 of the *EPHA4* gene. PCR products were cloned into the vector via the restriction sites BamHI and MluI for pCAS2, which is based on the pcDNA3.1 plasmid (Thermo Fisher Scientific, Waltham, MA, USA). Clones with wild-type or mutant genomic inserts were selected and verified by sequencing the cloned DNA fragments. The recombinant plasmids were transfected into 293T cells using Lipofectamine 3000 reagent (Thermo Fisher Scientific, Waltham, MA, USA). For RT-qPCR, total RNA was isolated from the transfected cells using Trizol reagent (Thermo Fisher Scientific, Waltham, MA, USA), and reverse transcription was performed using the GoScript Reverse Transcription System (Promega, Madison, MI, USA). PCR amplification was

performed using the pCAS2-RT-F and pCAS2-RT-R primers, and the products were sequenced using pCAS2-RT-F.

Primers sequences for the minigene assay were as follows:

pCAS2-RT-F: 5'-CTGACCCTGCTGACCCTCCT-3'
pCAS2-RT-R: 5'-TTGCTGAGAAGGCGTGGTAGAG-3'

## Nested PCR

The splicing variant (*EPHA4:* c.1318+10344A>G) was characterized by nested PCR. RNA was extracted from the whole blood of the patient using the TRIzon Reagent (CWBIO, Hangzhou, China) according to the manufacturer's guidelines. The cDNA was synthesized using HiScript II 1st Strand cDNA Synthesis Kit with a gDNA wiper (Vazyme, Nanjing, China) according to the manufacturer's guidelines. Nested PCR was performed as described previously (*Yao and Tavis, 2005*). Nested PCRs in 50 µl included 2 µl cDNA from the reverse transcription reaction as the template for the first round of PCR or 2 µl of first-round PCR product as the template for the second PCR, 1.5 µl 10 µM sense primer, 1.5 µl 10 µM anti-sense primer, 5 µl nucleotide mix (2 mM each dNTP), 5 µl 10× KOD Buffer, 1 µl 1 unit/µl Kod-Plus-Neo polymerase, 3 µl 25 mM $MgSO_4$ and 31 µl $ddH_2O$. The PCR program was (94°C for 2 min, 98°C for 10 s, 55°C for 30 s, 68°C for 60 s)×20 cycles and then (94°C for 2 min, 98°C for 10 s, 57°C for 30 s, 68°C for 60 s)×30 cycles. Primers sequences:

For the first round:

F: 5'-GGCTCCTGTGTCAACAACTC-3'
R: 5'-GTTGGGATCTTCGTACGTAA-3'

For the second round:

F: 5'-AACTGCCTATGCAACGCTGG-3'
R: 5'-AGCTGCAATGAGAATTACC-3'

## Western blots

This study utilized the HEK293T cell line, which was obtained from the Pricella Biotechnology Co. (catalog number CL-0005). The cell line was authenticated by short tandem repeat profiling. Mycoplasma testing confirmed that the HEK293T were free from contamination. HEK293T was selected for its high transfection efficiency to assess the impact of EPHA4 missense variant on protein expression. First, HEK293T cells were transfected with *EPHA4*-C1-pEGFP plasmid or the corresponding wild-type vector to detect the missense variant in *EPHA4* (c.2546G>A, p.Cys849Tyr). Cells were cultured in six-well plates and transfected with DNA (2 mg/well) using Lipofectamine 3000 reagent (Thermo Fisher Scientific, Waltham, MA, USA). After 48 hr, cells were harvested, and protein extracts were prepared as described (*Ding et al., 2017*). The mutant proteins and wild-type proteins were fused to GFP. The expression of the two proteins was compared by western blots with a GFP antibody using the ECL detection system. The GFP antibody was purchased from Cell Signaling Technology (Cell Signaling, Danvers, MA, USA). The CDK5 antibody was purchased from Santa Cruz Biotechnology (Santa Cruz, CA, USA). A phospho-specific antibody against CDK5 phosphorylated at Tyr (GeneTex, Irvine, CA, USA) was purchased from GeneTex. Western blot experiments were repeated twice with similar results for the replicates.

## Genotyping and imputation of GWAS

In the East Asian Japanese cohort, genotyping was performed by Illumina Human610 Genotyping BeadChip, Illumina HumanOmniExpressExome, and HumanOmniExpress as our previous GWASs (*Kou et al., 2019*; *Kou et al., 2013*; *Ogura et al., 2015*; *Takahashi et al., 2011*). For quality control (QC), subjects with a call rate<0.98, high-degree relatedness with other subjects, and outliers of East Asian ethnicity were excluded. For variants QC, the exclusion criteria of variants were as follows: call rate<0.99, p-value for Hardy-Weinberg equilibrium <1.0 × 10⁻⁶, and minor allele count <10. The reference panel for imputation, namely JEWEL7K, was composed of 1000 Genomes Project phase 3 (v5) (*Auton et al., 2015*) and Japanese whole-genome sequence data (*Das et al., 2016*) with 3256 high-depth subjects (≥30 read counts) and 4216 low-depth subjects (≤15 read counts). Using EAGLE

2.4.1 (https://alkesgroup.broadinstitute.org/Eagle/) to determine the haplotypes, pre-phasing was conducted. Genotypes were imputed using Minimac4 (v1.0.0) (*Das et al., 2016*). After imputation, we excluded variants with minor allele frequency (MAF) <0.005 and low imputation quality ($R^2 < 0.3$).

In the East Asian Hong Kong cohort, samples were genotyped with Illumina Human Omni Zhong-Hua-8 Beadchips. Illumina Genome Studio v2.0 was used to convert raw data into PLINK format. The QC steps of samples and variants were described in a previous study (*Marees et al., 2018*). Genotype phasing and imputing were executed using SHAPEIT v2.r900 (*Delaneau et al., 2012*) and IMPUTE2 (*Marchini et al., 2007*). The imputed data was filtered using the following parameters: INFO >0.6, Certainty >0.8, and MAF >0.01. Association analysis was performed using PLINK v1.9 logistic regression with covariates: sex, age, and top 20 principal components of the variance-standardized relationship matrix.

## East Asian GWAS meta-analysis for IS
For the meta-analysis of the four datasets (three Japanese datasets and one Hong Kong dataset), an inverse-variance-based method was performed by METAL (v2011-03-25) (*Willer et al., 2010*). SNPs in three or more of the four cohorts were used in subsequent analyses.

## Gene-based common variant analysis and eQTL analysis
SNPs in *EPHA4* or within 20 kb flanking *EPHA4* with a significant association with IS were retrieved. SNPs were matched to potential eQTLs according to the GTEx database (v8, https://gtexportal.org). Gene-based common variant analyses were performed using Multi-marker Analysis of GenoMic Annotation (MAGMA) (*de Leeuw et al., 2015*) and FUMA (*Watanabe et al., 2017*) using default settings with LD information from the 1000 Genomes Project East Asian population (1KGP EAS) as a reference. SNPs located within 2 kb upstream and 1 kb downstream from *EPHA4* were included in the gene-based analysis.

## Zebrafish strains, mutants, and morphants
Zebrafish Tuebingen (TU) strains were maintained at 28°C on a 14 hr/10 hr light/dark cycle. Embryos were raised at 28.5°C in E3 medium (5 mM NaCl, 0.17 mM KCl, 0.39 mM CaCl$_2$, 0.67 mM MgSO$_4$) following standard protocols. Zebrafish have two homologs of *EPHA4*, *epha4a* and *epha4b*. The CRISPR/Cas9 system was used to generate zebrafish *epha4a* and *epha4b* mutants. To increase efficiency, we injected Cas9 mRNA together with multiple single guide RNAs (sgRNAs) for each gene. The sgRNA sequences for *epha4a* and *epha4b* are listed in *Supplementary file 7*. Morpholino sequences for *epha4a* and *efnb3b* knockdown analysis are also listed in *Supplementary file 7*.

## Micro CT imaging
Adult zebrafish *epha4a* mutants or wild-type siblings were euthanized with tricaine methanesulfonate and fixed in 4% paraformaldehyde. Micro CT images were captured using a PerkinElmer Quantum GX2. Planar images acquired over 360° of rotation were reconstructed using QuantumGX. Three-dimensional renders of the skeleton were made with Analyze 12.0 software (AnalyzeDirect).

## Behavioral recordings and analysis
For behavior analysis, individual zebrafish larvae were transferred into a 24-well plate with fresh E3 medium at 8 dpf. Then, the plate was placed inside the Daniovision (Noldus) observation chamber for further behavior analysis. Video tracking of swimming activity and further statistical analysis were performed using the EthoVision XT10 software. Behavioral data were shown as total swim distance (mm), average velocity (mm/s), average relative turn angle (°), average absolute turn angle (degree), and average absolute angular velocity (°/s) at a duration for 4 min. A positive angle reflects a leftward turn.

For tactile stimulation, zebrafish larvae were placed in a concave slide, and tactile stimulation was performed with glass capillaries to the head or tail sides. Larval startle responses were recorded using a high-speed video camera (Mikrotron, EoSens Mini1) at 1000 fps. Automated analysis of larval movement was performed using the FLOTE software package (*Jain et al., 2014*).

For optogenetic studies, the Tg(Gal4$^{s1020t}$; *UAS*:ChR2) transgenic embryos were injected with control, *epha4a* or *efnb3b* morpholinos at one cell stage. At 5 dpf, the head of injected larva was

mounted in 1% low melting point agarose (Sigma), and the body was exposed in glass-bottom dishes (WPI). Leica M165FC fluorescence microscope was used to irradiate with a blue laser at 488 nm wavelength. Larval responses were recorded using a high-speed video camera (Mikrotron, EoSens Mini1) at 500 fps. Body curvature analysis of larval movement was performed using the FLOTE software package.

### Whole-mount in situ hybridization and immunofluorescence

The primer sequences used to amplify *epha4a*, *epha4b,* and *rfng* genes were listed in **Supplementary file 7**. Probe synthesis and whole-mount in situ hybridization were performed according to standard protocols. For RS neurons immunostaining, embryos were fixed in 2% trichloroacetic acid at 48 hpf for 3–4 hr, washed twice in 0.5% Triton X-100 in PBS and blocked in 0.5% Triton X-100, 10% normal goat serum, 0.1% bovine serum albumin (Solarbio) in PBS for 1 hr. The embryos were stained by monoclonal anti-neurofilament 160 antibody (Sigma-Aldrich) overnight at 4°C, then stained by goat anti-mouse Alexa Fluor 488 (Invitrogen) after washing.

### Analysis of neuronal calcium signals activity

Control, *epha4a* or *efnb3b* morpholinos were injected into Tg(*elavl3*:GAL4; UAS:GCaMP) transgenic embryos at one cell stage. At 24 hpf, embryos were paralyzed with 0.5 mg/ml α-bungarotoxin (Alomone Labs), then mounted in 1% low melting point agarose (Sigma) in glass-bottom dishes (WPI). Neuronal calcium signal images were collected with IXON-L-888 EMCCD camera equipped on Dragonfly 200 Spinning Disk Confocal Microscope using a ×20/0.55 objective within 1 min at a frame rate of 10 fps.

Images were analyzed with ImageJ software. To quantify the change in fluorescence intensity, a region of interest was defined, and the fluorescence intensity $F_t$ of different frames was normalized to $F_t/F_{min}$ based on the minimum fluorescence intensity $F_{min}$ in all frames. To compare the left-right alternation pattern of neuronal calcium signals, the ratio of the left-side calcium signals frequency ($f_{left}$) to the right-side calcium signals frequency ($f_{right}$) was logarithmically transformed (log-transformed).

### Quantification and statistical analysis

Statistical analyses were performed in SPSS (v15.0). Unpaired Student's t-test, Welch one-way ANOVA, or two-way ANOVA followed by Tukey's multiple comparison test was applied when appropriate. All experiments were replicated at least three times independently. $p < 0.05$ was considered statistically significant.

## Acknowledgements

We appreciate all the patients, their families, and clinical research coordinators, including physicians who participated in this project. The authors also acknowledge the Texas Advanced Computing Center (TACC) at The University of Texas at Austin for providing computing resources that have contributed to the results related to the TSRHC cohort. URL: http://www.tacc.utexas.edu. The Department of Molecular and Human Genetics at Baylor College of Medicine derives revenue from the chromosomal microarray analysis (CMA by aCGH and/or SNP arrays), clinical exome sequencing (cES), and whole-genome sequencing (WGS) offered in the Baylor Genetics (BG) Laboratory (http://bmgl.com).

## Additional information

### Competing interests

Pengfei Liu, Jennifer E Posey: affiliated to Baylor Genetics, Houston. DISCO study group (Deciphering Disorders Involving Scoliosis & COmorbidities): James R Lupski: J.R.L has stock ownership in 23andMe, is a paid consultant for Regeneron Pharmaceuticals and Novartis, and is a co-inventor on multiple United States and European patents related to molecular diagnostics for inherited neuropathies, eye diseases and bacterial genomic fingerprinting. The other authors declare that no competing interests exist.

## Funding

| Funder | Grant reference number | Author |
| --- | --- | --- |
| National Key Research and Development Program of China | 2023YFC2509700 | Lianlei Wang |
| National Natural Science Foundation of China | 81822030 | Nan Wu |
| National Natural Science Foundation of China | 82102522 | Lianlei Wang |
| National Natural Science Foundation of China | 31991194 & 32125015 | Chengtian Zhao |
| National Natural Science Foundation of China | 82172382 | Jianguo T Zhang |
| Chinese Academy of Medical Sciences Initiative for Innovative Medicine | 2021-I2M-1-051 | Nan Wu |
| Chinese Academy of Medical Sciences Initiative for Innovative Medicine | 2021-I2M-1-052 | Zhihong Wu |
| Chinese Academy of Medical Sciences Initiative for Innovative Medicine | 2020-I2M-C&T-B-030 | Jianguo T Zhang |
| Natural Science Foundation of Beijing Municipality | 7222133 | Shengru Wang |
| Non-profit Central Research Institute Fund of Chinese Academy of Medical Sciences | No. 2019PT320025 | Nan Wu |
| National High Level Hospital Clinical Research Funding | 2022-PUMCH-D-004 | Jianguo T Zhang |
| National High Level Hospital Clinical Research Funding | 2022-PUMCH-C-033 | Nan Wu |
| Natural Science Foundation of Shandong Province | ZR202102210113 | Lianlei Wang |
| Taishan Scholar Project of Shandong Province | | Lianlei Wang |
| Gabriella Miller Kids First Program grant | X01 HL132375-01A1 | Jonathan J Rios |

The funders had no role in study design, data collection and interpretation, or the decision to submit the work for publication.

## Author contributions

Lianlei Wang, Conceptualization, Resources, Data curation, Funding acquisition, Investigation, Methodology, Writing – original draft, Project administration; Xinyu Yang, Conceptualization, Data curation, Software, Formal analysis, Validation, Investigation, Methodology, Writing – original draft, Writing – review and editing; Sen Zhao, Conceptualization, Resources, Data curation, Software, Formal analysis, Supervision, Investigation, Visualization, Methodology, Writing – original draft, Project administration; Pengfei Zheng, Software, Formal analysis, Methodology, Writing – original draft; Wen Wen, Data curation, Software, Investigation; Kexin Xu, Data curation, Methodology, Writing – original draft; Xi Cheng, Data curation, Software, Formal analysis; Qing Li, Data curation, Software, Methodology; Anas M Khanshour, Yoshinao Koike, Resources, Data curation; Junjun Liu, Resources, Software; Xin Fan, Lulu Li, Investigation; Nao Otomo, Data curation, Software; Zefu Chen, Resources, Data curation,

Formal analysis, Methodology; Yaqi Li, Data curation; Haibo Xie, Conceptualization, Formal analysis, Supervision; Panpan Zhu, Software, Investigation; Xiaoxin Li, Resources, Investigation; Yuchen Niu, Guixing Qiu, Shiro Ikegawa, Supervision, Project administration; Shengru Wang, Jonathan J Rios, Supervision, Funding acquisition, Project administration; Sen Liu, Xiuli Zhao, Carol A Wise, Resources, Project administration; Suomao Yuan, Resources, Supervision; Chikashi Terao, Supervision, Methodology, Project administration; Ziquan Li, Jennifer E Posey, Resources; Shaoke Chen, Resources, Supervision, Project administration; Pengfei Liu, Data curation, Supervision, Methodology, Project administration; Zhihong Wu, Conceptualization, Resources; DISCO study group (Deciphering Disorders Involving Scoliosis & COmorbidities), Methodology, Project administration, Resources; James R Lupski, Resources, Supervision, Methodology, Project administration; Jianguo T Zhang, Resources, Supervision, Funding acquisition, Validation, Visualization, Writing – original draft, Project administration, Writing – review and editing; Chengtian Zhao, Conceptualization, Resources, Supervision, Funding acquisition, Validation, Visualization, Writing – original draft, Project administration, Writing – review and editing; Nan Wu, Conceptualization, Resources, Supervision, Funding acquisition, Validation, Visualization, Methodology, Writing – original draft, Project administration, Writing – review and editing

**Author ORCIDs**
Xinyu Yang http://orcid.org/0009-0002-6706-9857
Pengfei Zheng https://orcid.org/0000-0002-8729-0850
Wen Wen https://orcid.org/0000-0003-4858-8037
Yoshinao Koike https://orcid.org/0000-0002-5431-5572
Chikashi Terao https://orcid.org/0000-0002-6452-4095
Xiuli Zhao https://orcid.org/0000-0001-8238-6311
Jonathan J Rios https://orcid.org/0000-0002-0969-2184
Carol A Wise https://orcid.org/0000-0002-6790-2194
Chengtian Zhao https://orcid.org/0000-0003-1236-914X
Nan Wu https://orcid.org/0000-0002-9429-2889

**Ethics**
Human subjects: Approval for the study was obtained from the ethics committee at the Peking Union Medical College Hospital (JS-098, JS-2364), the medical ethics committee of the Keio University Hospital (No. 20080129), the ethical committee of RIKEN Yokohama Institute (No. H20-17(8)), and the Institutional Review Board of the University Texas Southwestern Medical Center (protocol STU 112010-150). Written informed consent was obtained from each participating individuals and families in the three cohorts. For the control group, the protocols were approved by the ethics committee at Peking Union Medical College Hospital.
Animal experimentation: All zebrafish studies were approved by the Animal Care Committee of the Ocean University of China (Animal protocol number: OUC2012316).

Reviewer #2 (Public review): https://doi.org/10.7554/eLife.95324.3.sa1
Author response https://doi.org/10.7554/eLife.95324.3.sa2

# Additional files

**Supplementary files**
Supplementary file 1. Summary of the 14 studies and their corresponding single-nucleotide polymorphisms (SNPs) included in the candidate genes mapping.

Supplementary file 2. Summarized results of burden analysis.

Supplementary file 3. The significant single-nucleotide polymorphisms (SNPs) in the EPHA4 region identified from East Asian genome-wide association study (GWAS) meta-analysis.

Supplementary file 4. Search strategies for each database.

Supplementary file 5. Inclusion and exclusion criteria of literature review.

Supplementary file 6. Sequencing information of PUMCH idiopathic scoliosis (IS) cohort.

Supplementary file 7. Sequences information of single guide RNA (sgRNA) and primers for zebrafish study.

MDAR checklist

## Data availability

The raw whole exome/genome sequencing data from this study have been uploaded in the Genome Sequence Archive in BIG Data Center (https://bigd.big.ac.cn/), Beijing Institute of Genomics (BIG), Chinese Academy of Sciences (Accession Number: HRA011763, HRA006052). Other datasets used and/or analyzed during the current study are described in supplementary files, figure supplements, and the section of "Materials and Methods".

The following datasets were generated:

| Author(s) | Year | Dataset title | Dataset URL | Database and Identifier |
|---|---|---|---|---|
| Wu N, Zhang J, Wu Z | 2025 | EPHA4 signaling dysregulation links abnormal locomotion and the development of idiopathic scoliosis | https://ngdc.cncb.ac.cn/gsa-human/browse/HRA011763 | National Genomics Data center, HRA011763 |
| Wu N, Zhang J, Wu Z | 2023 | control sequencing data | https://ngdc.cncb.ac.cn/gsa-human/browse/HRA006052 | National Genomics Data center, HRA006052 |

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
