## [Editor Report · eLife Assessment]

Genetic variants have been strongly implicated in idiopathic scoliosis (IS), however, the list of variants that are causative of IS is not complete and the cellular and molecular mechanisms that underlie IS are poorly understood. These authors combined human genetic analysis with zebrafish experiments to produce **valuable** evidence that alleles that impair function of EPHA4 cause IS, thereby extending our understanding of the basis of IS. The human genetic data are quite **convincing** but the zebrafish work lacks some validations and details.

---

## [Referee Report · Reviewer #2 (Public review)]

Summary:

Idiopathic scoliosis (IS) is a common spinal deformity. Various studies have linked genes to IS, but underlying mechanisms are unclear such that we still lack understanding of the causes of IS. The current manuscript analyzes IS patient populations and identified EPHA4 as a novel associated gene, finding three rare variants in EPHA4 from three patients (one disrupting splicing and two missense variants) as well as a large deletion (encompassing EPHA4) in a Waardenburg syndrome patient with scoliosis. EPHA4 is a member of the Eph receptor family. Drawing on data from zebrafish experiments, the authors argue that EPHA4 loss of function disrupts central pattern generator (CPG) function necessary for motor coordination.

Strengths:

The main strength of this manuscript is the human genetic data, which provide convincing evidence linking EPHA4 variants to IS. The loss of function experiments in zebrafish strongly support the conclusion that EPHA4 variants that reduce function lead to IS.

Weaknesses:

The conclusion that disruption of CPG function causes spinal curves in the zebrafish model is not fully supported. The authors' final model is that a disrupted CPG leads to asymmetric mechanical loading on the spine and, over time, the development of curves. This is a reasonable idea, but currently not strongly backed up by data in the manuscript. Potentially, the impaired larval movements simply coincide with, but do not cause, juvenile-onset scoliosis. Support for the authors' conclusion would require independent methods of disrupting CPG function and determining if this is accompanied by spine curvature. Nevertheless, the data showing correlations between spine curvature and abnormal neural patterning, neuronal firing, and swimming in eph4a loss-of-function mutant larvae are sound.

Comments on revisions:

I think the authors misunderstood my point about genetic nomenclature for the zebrafish alleles. The nomenclature guidelines are described at ZFIN, and ZFIN will ask for appropriate allele designations.

---

## [Author Response]

The following is the authors’ response to the original reviews

**Joint Public Review:**
Idiopathic scoliosis (IS) is a common spinal deformity. Various studies have linked genes to IS, but underlying mechanisms are unclear such that we still lack understanding of the causes of IS. The current manuscript analyzes IS patient populations and identifies EPHA4 as a novel associated gene, finding three rare variants in EPHA4 from three patients (one disrupting splicing and two missense variants) as well as a large deletion (encompassing EPHA4) in a Waardenburg syndrome patient with scoliosis. EPHA4 is a member of the Eph receptor family. Drawing on data from zebrafish experiments, the authors argue that EPHA4 loss of function disrupts the central pattern generator (CPG) function necessary for motor coordination.The main strength of this manuscript is the human genetic data, which provides convincing evidence linking EPHA4 variants to IS. The loss of function experiments in zebrafish strongly support the conclusion that EPHA4 variants that reduce function lead to IS.The conclusion that disruption of CPG function causes spinal curves in the zebrafish model is not well supported. The authors' final model is that a disrupted CPG leads to asymmetric mechanical loading on the spine and, over time, the development of curves. This is a reasonable idea, but currently not strongly backed up by data in the manuscript. Potentially, the impaired larval movements simply coincide with, but do not cause, juvenile-onset scoliosis. Support for the authors' conclusion would require independent methods of disrupting CPG function and determining if this is accompanied by spine curvature. At a minimum, the language of the manuscript could be toned down, with the CPG defects put forward as a potential explanation for scoliosis in the discussion rather than as something this manuscript has "shown". An additional weakness of the manuscript is that the zebrafish genetic tools are not sufficiently validated to provide full confidence in the data and conclusions.

We highly appreciate the reviewer’s insightful comments and the acknowledgment of the main values of our study. We agree with the reviewer that further experiments are needed to fully establish the relationship between CPG and scoliosis. In response, we have revised the conclusion in the manuscript to better reflect this. Additionally, we conducted further analyses on the mutants to provide additional evidence supporting this concept.

**Reviewer #1 (Recommendations for the authors):**
Epha4a mutant zebrafish exhibited mild spinal curves, mostly laterally and in the tail. This was 75% of homozyous mutants but also, surprisingly, about 20% of heterozygotes. epha4b mutants also developed some mild scoliosis. If the two zebrafish paralogs can compensate for each other (partial redundancy), we might expect more severe scoliosis in double mutants. Did the authors generate and analyze double mutants? I believe it would be very useful for this study to report the zebrafish phenotype of loss of both paralogs together.

We appreciate the reviewer’s insightful comment regarding the potential value of reporting the phenotype of *eph4a/eph4b* double mutants. While we fully agree that this analysis would be valuable, our attempts to generate double mutants have been unsuccessful. These two genes are closely linked on the chromosome, with less than 100 kb separating them, which makes it challenging to generate double mutants through standard genetic crossing. Establishing a double mutant line would require more than a year due to the technical constraints of the process. Although we are unable to address this question directly at this time, we hypothesize that *eph4a/eph4b* double mutants may exhibit a higher likelihood of body axis abnormalities based on the phenotypes observed in single mutants and the known functions of these genes.

We hope this perspective will provide some useful context despite the limitations.

In Figure 1F, a pCDK5 western blot is performed as a readout of EPH4A signaling after either WT or C849Y mutant EPH4A is transfected into HEK 293T cells. It would be useful to mention in the text, or at least the figure legend, how this experiment was performed/where the protein samples came from. It is included in the methods, but in the main text, it simply says "we conducted western blotting" without mentioning whether the protein samples were from cell lines, patients, or another source.

Sorry for our ignorance. A detailed description of the western blotting conduction was supplemented at both “results” part (page 8, line 187-190) and the Figure 1 legend.

Was the relative turn angle biased to the left or right side of the fish? (i.e. is a positive angle a rightward or leftward turn?)

We are sorry for our unclear description. In Figure 3D, positive angle means turning left, while negative angle means turning right. In wild-type larvae, the average turning angle over a 4-minute period is approximately 0, whereas in mutants, this value deviates from 0, indicating a directional preference (positive for leftward and negative for rightward turns) in swimming behavior during the recording period. We have also made the necessary supplementation in the text and figure legend.

In Figure 4, morpholinos rather than mutants are used, but it is not clear why. Has it been established that the MO used disrupts gene function specifically? Can the effect of the MO be rescued by expressing a wild-type mRNA of Epha4a? Does MO knockdown induce spinal curves if fish are raised? Indeed, this could be a way to determine whether the spinal curves are caused by early events in development (when MOs are active).

Thanks for the comments. The efficacy of relevant MOs has been well-documented in numerous previous studies (Addison et al., 2018; Cavodeassi et al., 2013; Letelier et al., 2018; Royet et al., 2017). Following this reviewer’s suggestion, we have raised the *epha4a* morphants into adults, while no scoliosis were observed, suggesting that the spinal curvature formation may be induced by long-term defects in the absence of Epha4a. Additionally, we reconfirmed the abnormal motor neuron activation frequency phenotype in the mutants background. The corresponding data have replaced the original Figure 4 in the manuscript.

References

(1) Addison, M., Xu, Q., Cayuso, J., and Wilkinson, D.G. (2018). Cell Identity Switching Regulated by Retinoic Acid Signaling Maintains Homogeneous Segments in the Hindbrain. Dev Cell 45, 606-620 e603.

(2) Cavodeassi, F., Ivanovitch, K., and Wilson, S.W. (2013). Eph/Ephrin signalling maintains eye field segregation from adjacent neural plate territories during forebrain morphogenesis. Development 140, 4193-4202.

(3) Letelier, J., Terriente, J., Belzunce, I., Voltes, A., Undurraga, C.A., Polvillo, R., Devos, L., Tena, J.J., Maeso, I., Retaux, S., et al. (2018). Evolutionary emergence of the rac3b/rfng/sgca regulatory cluster refined mechanisms for hindbrain boundaries formation. Proc Natl Acad Sci U S A 115, E3731-E3740.

(4) Royet, A., Broutier, L., Coissieux, M.M., Malleval, C., Gadot, N., Maillet, D., Gratadou-Hupon, L., Bernet, A., Nony, P., Treilleux, I., et al. (2017). Ephrin-B3 supports glioblastoma growth by inhibiting apoptosis induced by the dependence receptor EphA4. Oncotarget 8, 23750-23759.

**Reviewer #2 (Recommendations for the authors):**
Supplementary Table 3 is missing.

Sorry for any inconvenience caused to the reviewers. Due to the size of the supplementary Table 3, we have separately uploaded an Excel file as supplementary materials. We have also double-checked during the resubmission process of the revised manuscript. Thanks for your thorough review.

The authors report only a single mutant allele for zebrafish epha4a and epha4b. Additionally, they provide no information about how many generations each allele has been outcrossed. The authors should provide some type of validation that the phenotypes they describe result from loss of function of the targeted gene and not from an off-targeting event.

Thanks for the comments. For *epha4a* and *epha4b* mutants, each homozygous mutant was initially derived from the self-crossing of first filial generation heterozygotes, and subsequent homozygous generations were maintained for fewer than three rounds of in-crossing. Interestingly, we observed a reduction in the incidence of scoliosis across successive generations. This trend may be attributed to potential genetic compensation mechanisms, which could mitigate the phenotypic severity over time. To address concerns about possible off-target effects, we synthesized and injected *epha4a* mRNA to test for phenotypic rescue. Our data show that *epha4a* mRNA injection partially restored swimming coordination in the mutants (Fig. S5). Moreover, similar motor coordination defects have been reported in Epha4-deficient mice, as documented in previous studies (Kullander et al., 2003; Borgius et al., 2014). These findings collectively strengthen the hypothesis that Epha4a plays a critical role in regulating motor coordination.

References

(1) Borgius, L., Nishimaru, H., Caldeira, V., Kunugise, Y., Low, P., Reig, R., Itohara, S., Iwasato, T., and Kiehn, O. (2014). Spinal glutamatergic neurons defined by EphA4 signaling are essential components of normal locomotor circuits. J Neurosci 34, 3841-3853.

(2) Kullander, K., Butt, S.J., Lebret, J.M., Lundfald, L., Restrepo, C.E., Rydstrom, A., Klein, R., and Kiehn, O. (2003). Role of EphA4 and EphrinB3 in local neuronal circuits that control walking. Science 299, 1889-1892.

The authors need to provide allele designations for the mutant alleles following accepted nomenclature guidelines.

Thank you for your careful review! We have reviewed and made revisions to the genes and mutation symbols throughout the entire text.

The three antisense morpholino oligonucleotides need to be validated for efficacy and specificity.

Thanks for the comments. The morpholinos were extensively used and validated in previous studies, and the efficacy of these morpholinos has been thoroughly validated in multiple studies (Addison et al., 2018; Cavodeassi et al., 2013; Letelier et al., 2018; Royet et al., 2017). Furthermore, we also performed swimming behavior analysis in the mutant background, which showed similar results as the morphants. Moreover, we also performed rescue experiments to confirm the specificity of the mutants (Fig. S5). Finally, we reconfirmed the abnormal calcium signaling in the mutants (Fig. 4), which further support our previous knockdown results.

References

(1) Addison, M., Xu, Q., Cayuso, J., and Wilkinson, D.G. (2018). Cell Identity Switching Regulated by Retinoic Acid Signaling Maintains Homogeneous Segments in the Hindbrain. Dev Cell 45, 606-620 e603.

(2) Cavodeassi, F., Ivanovitch, K., and Wilson, S.W. (2013). Eph/Ephrin signalling maintains eye field segregation from adjacent neural plate territories during forebrain morphogenesis. Development 140, 4193-4202.

(3) Letelier, J., Terriente, J., Belzunce, I., Voltes, A., Undurraga, C.A., Polvillo, R., Devos, L., Tena, J.J., Maeso, I., Retaux, S., et al. (2018). Evolutionary emergence of the rac3b/rfng/sgca regulatory cluster refined mechanisms for hindbrain boundaries formation. Proc Natl Acad Sci U S A 115, E3731-E3740.

(4) Royet, A., Broutier, L., Coissieux, M.M., Malleval, C., Gadot, N., Maillet, D., Gratadou-Hupon, L., Bernet, A., Nony, P., Treilleux, I., et al. (2017). Ephrin-B3 supports glioblastoma growth by inhibiting apoptosis induced by the dependence receptor EphA4. Oncotarget 8, 23750-23759.

Line 229. "While in consistent with previous reports, the hindbrain rhombomeric boundaries were found to be defective....". This sentence is not clear. Please describe how it is "inconsistent".

Thanks for the comments and sorry for the unclear description, we have described this more clearly in our revised manuscript (page 9, line 229-230).

Animals frequently are described as "heterozygous mutants" or "mutants". Please make clear that the latter are homozygous mutant animals.

Thanks for the comments. In the manuscript, all references to mutants specifically indicate homozygous mutants. Heterozygous mutants are explicitly identified as such.

The chromatin interaction portion of the Methods does not include any information on how these experiments were conducted or where the data were obtained. This information needs to be provided.

Thanks for your advice. The detailed information of chromatin interaction mapping has been provided in “Methods and Materials” (page 18-19, line 450-455). Information about the interacting regions was derived from Hi-C datasets of 21 tissues and cell types provided by GSE87112. The significance of interactions for Hi-C datasets was computed by Fit-Hi-C, with an FDR ≤ 10-6 considered significant.

The authors present single-cell RNA-seq data in Supplementary Figure 5 for which they cite Cavone et al, 2021. This seems like an odd database to use. Can the authors provide an explanation for choosing it? In any case, the citation should also be made in the Supplementary Figure 5 legend.

Thank you for your rigorous comment, we have cited this literature in the proper place of the revised manuscript. Cavone et al. used the *her4.3*:GFP line to label ependymo-radial glia (ERG) progenitor cells and performed single-cell RNA-seq on FACS-isolated fluorescent cells. The isolated cells included not only ERG progenitors but also undifferentiated and differentiated neurons and oligodendrocytes. The authors attributed this to the relative stability of the GFP protein, which remained in the progeny of GFP-expressing *her4.3*+ ERG progenitor cells, thus effectively acting as a short-term cell lineage tracer. Indeed, clustering analysis of this data successfully identifies neural progenitors and other neural clusters. Therefore, we consider that this scRNA-seq data encompasses a comprehensive range of neural cell types and is suitable for analyzing the expression of genes of interest. Furthermore, we downloaded and analyzed the scRNA-seq data of the zebrafish nervous system reported by Scott et al. in 2021 (Fig. S7B) (Scott et al., 2021). Despite differences in the developmental stages of the larvae analyzed (Cavone et al. examined larvae at 4 dpf, whereas Scott et al. analyzed larvae at 24, 36, and 48 hpf), our findings are consistent. Specifically, *epha4a* and *epha4b* are expressed in interneurons, whereas *efnb3a* and *efnb3b* are enriched in floor plate cells.

References

(1) Scott, K., O'Rourke, R., Winkler, C.C., Kearns, C.A., and Appel, B. (2021). Temporal single-cell transcriptomes of zebrafish spinal cord pMN progenitors reveal distinct neuronal and glial progenitor populations. Dev Biol 479, 37-50.

In Figure Legend 1, "expressed from the EPHA4-mutant plasmid" is not an accurate description of the experiment.

Sorry for the previous inaccurate description. The description has been revised to accurately reflect the experiment. “Western blot analysis of *EPHA4*-c.2546G>A variant showing the protein expression levels of EPHA4 and CDK5 and the amount of phosphorylated CDK5 (pCDK5) in HEK293T cells transfected with *EPHA4*-mutant or *EPHA4*-WT plasmid”.

Figure 3 panels J and K need more explanation. I don't understand what the different colors represent nor do I understand what are wild type and what are mutant data.

Thank you for your valuable feedback. We apologize for the lack of clarity in the original figure legend. To address this, we have revised the legend of Figure 3 to provide a more detailed explanation. In panels J and K, each color-coded curve represents the response of an individual larva from an independent experimental trial to the stimulus. Specifically, panel J depicts the response data for the wild-type larvae, whereas panel K presents the response data for the homozygous *epha4a* mutants.

Please provide the genotypes for the images in Figure 5A.

Thanks for the comments and we are sorry for our unclear description, we have described this more clearly in the Figure 5.

Figure legend 6B should also note the heterozygote data with the wild type and homozygous mutant data.

Thanks for the comments, the data are now included in Figure 6B.

Epha4 and Efnb3 have well-established roles in axon guidance. Although this is noted in the Discussion, I think a more extensive description of prior findings would be helpful.

Thanks for your valuable feedback. A more detailed description of the roles of Epha4 and Efnb3 in axon guidance was provided in the “Discussion” (page 16, line 388-396).

The main conclusion of this manuscript is that EPHA4 variants cause IS by disrupting central pattern generator function. I think this is misleading. I think that the more valid conclusion is that EPHA4 loss of function causes axon pathfinding defects that impair locomotion by disrupting CPG activity, thereby leading to IS. I urge the authors to consider this more nuanced interpretation.

Thank you for your insightful comments. We appreciate your suggestion to refine our main conclusion. We agree that the proposed revision more accurately reflects our findings and will revise the manuscript accordingly to state that “EPHA4 loss of function causes axon pathfinding defects, which impair locomotion by disrupting central pattern generator activity, potentially leading to IS.”